# Temporal identity establishes columnar neuron morphology, connectivity, and function in a *Drosophila* navigation circuit

Luis F Sullivan[1,2], Timothy L Warren[1,2], Chris Q Doe[2]*

[1]Institute of Neuroscience, Howard Hughes Medical Institute, University of Oregon, Eugene, United States; [2]Institute of Molecular Biology, Howard Hughes Medical Institute, University of Oregon, Eugene, United States

**Abstract** The insect central complex (CX) is a conserved brain region containing 60 + neuronal subtypes, several of which contribute to navigation. It is not known how CX neuronal diversity is generated or how developmental origin of subtypes relates to function. We mapped the developmental origin of four key CX subtypes and found that neurons with similar origin have similar axon/dendrite targeting. Moreover, we found that the temporal transcription factor (TTF) Eyeless/Pax6 regulates the development of two recurrently-connected CX subtypes: Eyeless loss simultaneously produces ectopic P-EN neurons with normal axon/dendrite projections, and reduces the number of E-PG neurons. Furthermore, transient loss of Eyeless during development impairs adult flies' capacity to perform celestial navigation. We conclude that neurons with similar developmental origin have similar connectivity, that Eyeless maintains equal E-PG and P-EN neuron number, and that Eyeless is required for the development of circuits that control adult navigation.
DOI: https://doi.org/10.7554/eLife.43482.001

*For correspondence:
cdoe@uoregon.edu

**Competing interests:** The authors declare that no competing interests exist.

## Introduction

Work over the past two decades has revealed two important developmental mechanisms that generate neuronal diversity from flies to mice. First, spatial patterning cues produce different pools of neural progenitors (called neuroblasts in insects); second, neuronal progenitors/neuroblasts sequentially express a series of transcription factors that generate additional neuronal diversity (*Kohwi and Doe, 2013*). These so-called 'temporal transcription factors' or TTFs are expressed transiently in progenitors, are inherited by neurons born during the expression window, and specify progenitor-specific neuronal identity (*Rossi et al., 2017*; *Doe, 2017*). For example, the Hunchback (Hb) TTF is present in *Drosophila* embryonic neuroblasts as they produce their first progeny; loss of Hb leads to absence of first-born neurons, whereas prolonging Hb expression generates ectopic first-born neurons (*Isshiki et al., 2001*). While TTFs are clearly important for generating molecularly distinct neuronal subtypes, their role in establishing neuronal morphology, connectivity, and behavior remains relatively poorly understood.

Recent work has shown that there are only four bilateral 'type II' neuroblasts that generate the intrinsic neurons of the central complex (CX) projecting to the protocerebral bridge (PB). These four neuroblasts are named DM1-DM4 (*Yang et al., 2013*; *Andrade et al., 2019*) or DM1-DM3 and DM6 (*Riebli et al., 2013*); here we use the DM1-DM4 nomenclature (*Figure 1A*). Type II neuroblasts have a complex lineage. They repeatedly divide every 1.6 hr to generate a series of molecularly distinct intermediate neural progenitors (INPs), which in turn divide every 2–3 hr to produce 4–6 molecularly distinct ganglion mother cells (GMCs) that each yield a pair of sibling neurons (*Figure 1B*) (*Bello et al., 2008*; *Boone and Doe, 2008*; *Bowman et al., 2008*; *Homem et al., 2013*). Several laboratories have identified candidate temporal transcription factors (TTFs) that are expressed in type II

**eLife digest** Every task that an animal performs, even a simple one, typically requires numerous signals to pass across complex networks of cells called neurons. These networks develop early in an animal's life, beginning when progenitor cells called neural stem cells divide over and over to produce new cells. Specific molecular signals then induce these new cells to become different types of neurons. However, in many animals, it is poorly understood what these critical molecular signals are and how they work.

Fruit flies, for example, have a network of neurons that control how they navigate when flying. The same type of progenitor cell gives rise to at least four types of neurons in this network; these progenitor cells make an increasing amount of a protein called Eyeless as they age.

Sullivan et al. have now specifically disrupted production of the Eyeless protein in the progenitor cells, and found that this altered the relative numbers of navigation neurons. The fruit flies had too many of some types of navigation neurons and too few of others. Fruit flies normally navigate in a variety of directions relative to the sun, which may allow them to disperse and find food. This was not the case in experiments where the production of Eyeless was briefly disrupted when the flies were larvae. In these experiments, the adult flies tended to head towards a bright light (that represented the sun) much more often than normal, which would presumably keep them from dispersing effectively. This was true even if the disruption of Eyeless was not long enough to change the numbers of neuron types, showing the protein is important in determining both how these navigation neurons form networks, and whether they are born at all.

A better understanding of the complexities of how healthy networks of neurons develop may give scientists more insight into what goes wrong during human developmental disorders that affect the brain. In theory, it may also someday lead to tools that can help to repair the brain if it is damaged.

DOI: https://doi.org/10.7554/eLife.43482.002

neuroblasts, such as the Ecdysone Receptor (EcR) (*Figure 1B*, horizontal axis; *Syed et al., 2017*) or in INPs, such as Dichaete and Eyeless (*Figure 1B*, vertical axis; *Bayraktar and Doe, 2013*). Each of these TTFs is required to specify the identity of neurons born during its neuroblast or INP expression window (*Bayraktar and Doe, 2013*; *Ren et al., 2017*; *Syed et al., 2017*).

In this study we address how larval brain TTFs contribute to the development and function of the adult insect central complex (CX). The CX is a highly conserved brain region in insects that is thought to play a crucial role in navigation and motor control (*Pfeiffer and Homberg, 2014*; *Green et al., 2017*; *Heinze, 2017*; *Kim et al., 2017*; *Stone et al., 2017*; *Franconville et al., 2018*; *Giraldo et al., 2018*; *Green et al., 2018*). The CX is characterized by four distinct neuropil regions: the Ellipsoid Body (EB), Fan-shaped Body (FB), Protocerebral Bridge (PB), and Noduli (NO); the CX is also connected to lateral neuropils termed the Gall and the Round body (ROB) (*Wolff et al., 2015*). Columnar neurons, which innervate single glomeruli that tile the entire EB and PB neuropil, have been shown to play a key role in navigation (*Pfeiffer and Homberg, 2014*; *Green et al., 2017*; *Heinze, 2017*; *Kim et al., 2017*; *Turner-Evans et al., 2017*; *Franconville et al., 2018*; *Giraldo et al., 2018*; *Green et al., 2018*). There are at least four columnar neuron subtypes (*Figure 1C*). The E-PG neurons have spiny dendritic arbors in the EB (hence the E at the front of their name) and provide outputs to the PB and Gall (hence the PG at the end of their name); conversely, P-EN neurons have spiny dendritic arbors in the PB and provide outputs to the EB and Noduli. Recently it has been proposed that the E-PG/P-EN neurons form a recurrent circuit that tracks the fly's orientation in space (*Lin et al., 2013*; *Green et al., 2017*; *Turner-Evans et al., 2017*; *Green et al., 2018*). Two additional columnar neuron classes are PF-R neurons that have dendritic spines in the PB and FB and project axons to the ROB, and the P-FN neurons which have dendritic spines in the PB and project axons to the FB and Noduli (*Figure 1A*) (*Wolff et al., 2015*; *Wolff and Rubin, 2018*); both are proposed to have a role in navigation based on anatomical connectivity (*Heinze, 2017*; *Stone et al., 2017*; *Wolff and Rubin, 2018*), but their function has not been experimentally determined.

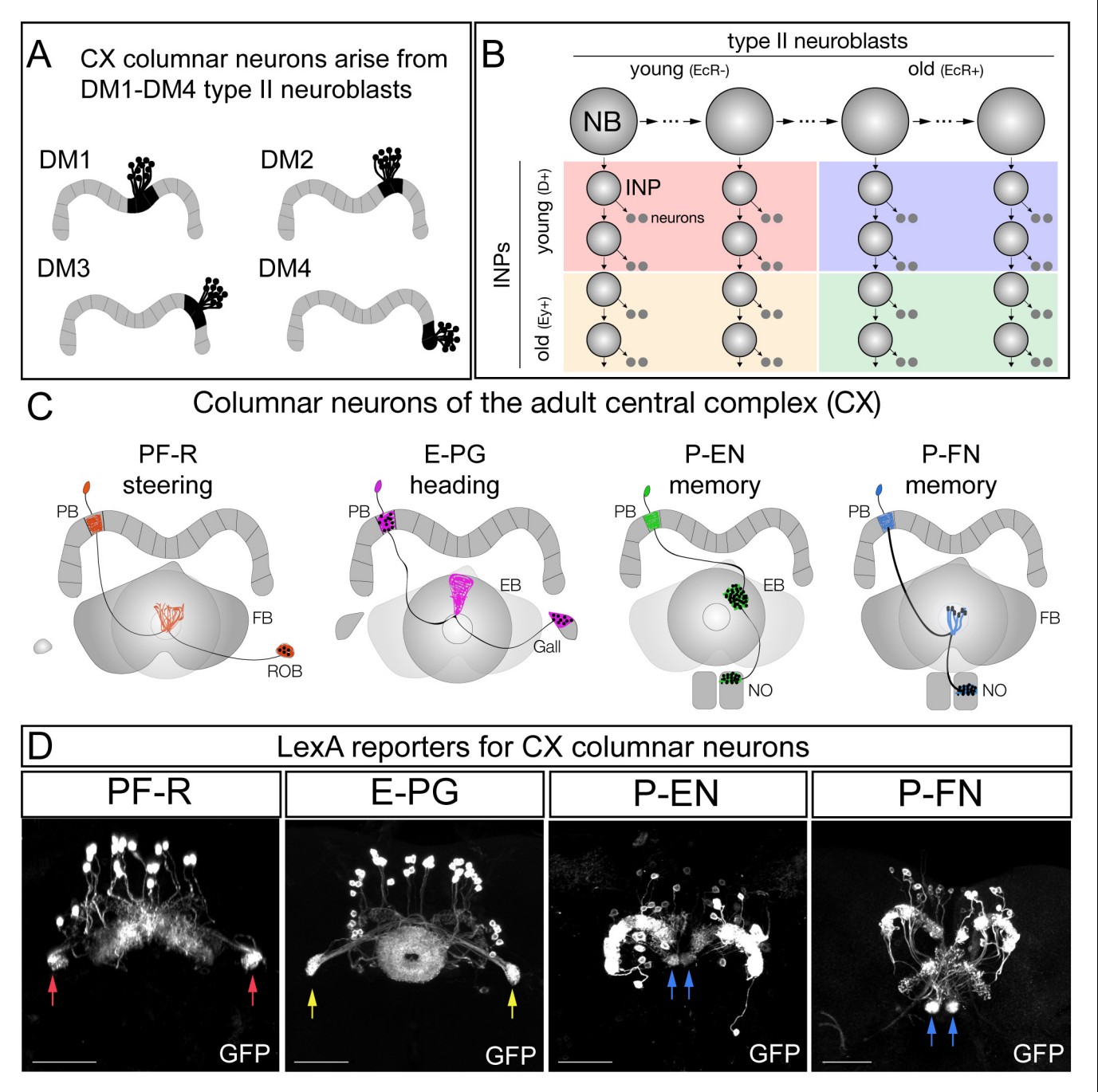

**Figure 1.** CX columnar neurons are generated by type II neuroblast lineages. (**A**) CX columnar neurons innervating the PB originate specifically from each of four bilateral type II neuroblast lineages (DM1-DM4), which include all four neuronal subtypes shown in panel D. DM1 lineage neurons innervate the most medial PB glomeruli, and DM4 lineage neurons innervate the most lateral PB glomeruli. Adult brain right hemisphere shown. (**B**) Type II neuroblasts divide every 1.6 hr to generate ~60 INPs; each INP progeny divides every 2–3 hr to produce 10–12 neurons (*Homem et al., 2013*). Both neuroblasts and INPs express temporal transcription factors that subdivide their lineages into distinct molecular windows. Finer subdivisions exist but are not shown for clarity. (**C**) PF-R, E-PG, P-EN, and P-FN columnar neuron subtypes; each has a proposed function in navigation (*Stone et al., 2017*; ) and a distinct pattern of connectivity. PB, protocerebral bridge; FB, fan-shaped body; ROB, round body; EB, ellipsoid body; NO, noduli. (**D**) Adult CX columnar neurons derived from INPs labeled with adult LexA lines specific for each subtype; see *Figure 1—figure supplement 1A* for genetic details. ROB, red arrows; Gall, yellow arrows; Noduli, blue arrows. Scale bars, 40μm. Genotypes: PF-R, *UAS-FLP; R9D11-Gal4, R37G12-lexA; lexAop(FRT.stop) mCD8::GFP*; E-PG, *UAS-FLP; R9D11-Gal4, R60D05-lexA; lexAop(FRT.stop)mCD8::GFP*; P-EN, *UAS-FLP; R9D11-Gal4, R12D09-lexA; lexAop(FRT.stop) mCD8::GFP*; P-FN, *UAS-FLP; R9D11-Gal4, R16D01-lexA; lexAop(FRT.stop)mCD8::GFP.*

*Figure 1 continued on next page*

*Figure 1 continued*

DOI: https://doi.org/10.7554/eLife.43482.003

The following figure supplement is available for figure 1:

**Figure supplement 1.** Intersectional genetic birthdating schemes.

DOI: https://doi.org/10.7554/eLife.43482.004

Here we map the developmental origin of these four CX neuronal subtypes postulated to have a critical role in navigation. We find that each is derived from a specific temporal window during the INP cell lineage, and that neurons with similar developmental origins have similar axon/dendrite neuropil targets. We confirm that Eyeless, previously shown to be a INP TTF (*Bayraktar and Doe, 2013*), is expressed in the latter half of INP lineages; we go on to show that Eyeless is required to promote the identity of the two CX neuron subtypes born late in INP lineages (E-PG, PF-R) as well as to repress the identity of the two CX neuron subtypes born during early INP lineages (P-EN, P-FN). In this way, the Eyeless TTF regulates the relative proportion of each neuronal subtype: loss of Eyeless generates fewer E-PG neurons and more P-EN neurons. Importantly, the ectopic P-EN neurons have normal anatomical connectivity. Finally, we show that loss of Eyeless specifically during the larval stages when E-PG neurons are born results in a highly specific defect in adult flight navigation, consistent with the proposed role of E-PGs in maintaining an arbitrary heading to a sun stimulus. Our findings are the first to identify the developmental origin of functionally important adult flight navigation neurons. Moreover, they set the stage for manipulating developmental genetic programs to alter the number and function of each class of adult CX neurons.

## Results

### CX columnar neurons are generated by type II neuroblast lineages

We used intersectional genetics to map the developmental origin of four CX columnar types (*Figure 1—figure supplement 1*). Our strategy was to use the FLP enzyme to permanently open a *lexAop-FRT-stop-FRT-GFP* reporter in specific populations of INPs and then use adult columnar neuron LexA transgenes to determine the number of each adult columnar neuron type made by each of these INP populations. This approach allowed us to map the developmental origin of neurons labeled by LexA reporters only at pupal or adult stages. We opened the *lexAop-FRT-stop-FRT-GFP* reporter in all INPs of the type II neuroblast lineages and confirmed that all four types of adult CX columnar neurons are generated by type II neuroblasts (*Figure 1—figure supplement 1A*). Indeed, we found that type II neuroblasts make all 30 PF-R neurons, all 40 E-PG neurons, all 40 P-EN neurons, and all 50 P-FN neurons across both hemispheres of the adult brain (*Figure 1D*). We conclude that the four types of CX columnar neurons are all derived from type II neuroblast lineages.

### CX columnar neurons are generated by young type II neuroblast lineages

The challenge in birth-dating CX neurons from type II neuroblast lineages is that they are generated across two temporal axes, NB and INP. To address this, we systematically dissected one axis at a time. Larval type II neuroblasts produce neurons over five days (0–120 hr after larval hatching; ALH), with each lineage generating roughly between 40–50 INPs, totaling around 400 neurons and additional glia from each distinct lineage (*Homem et al., 2013*). We used intersectional genetics to determine when each columnar neuron subtype was born during the type II neuroblast lineage. We transiently expressed the FLP recombinase in INPs to permanently open the lexAop reporter at different times during type II neuroblast lineages and assayed for the number of PF-R, E-PG, P-EN, or P-FN adult neurons made at each time-point (method summarized in *Figure 1—figure supplement 1B*). We found that PF-R neurons were made first in larval type II neuroblast lineages, followed by E-PG neurons, and then by P-EN and P-FN neurons which share overlapping birthdates (*Figure 2A*). The relatively broad distribution of columnar neuron birthdates is likely due to DM1-DM4 individual lineages generating neuron subtypes asynchronously, but could also represent natural developmental variation or stochasticity in the time of columnar neuron birthdates; it is most consistent with

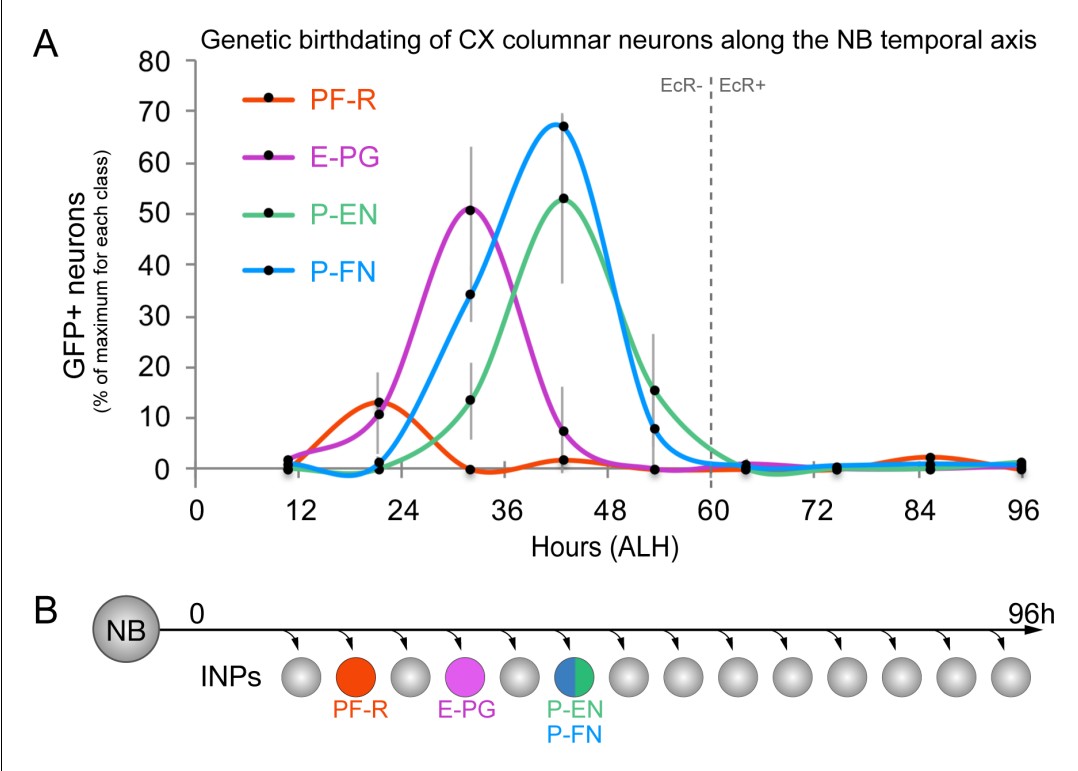

**Figure 2.** CX columnar neurons are generated by young type II neuroblast lineages. (**A**) Identifying the time during the neuroblast lineage that produces each columnar neuron subtype. See *Figure 1—figure supplement 1B* for genetic details. Note that PF-R neurons are born first, E-PG neurons second, and then P-EN/P-FN neurons sharing a common birthdate (n = 3–6 per time-point). (**B**) Summary of NB birthdating results.
DOI: https://doi.org/10.7554/eLife.43482.005

each pool of 30–50 columnar neurons being generated within a 12 hr temporal window in the type II neuroblast lineage (*Figure 2B*).

## CX columnar neurons with similar developmental origin have similar axon/dendrite targeting

We next defined columnar neuron birthdates along the INP temporal axis (see *Figure 1—figure supplement 1C*). Young INPs express Sox family transcription factor Dichaete (D), whereas old INPs express the Pax6 family transcription factor Eyeless (*Bayraktar and Doe, 2013*; *Eroglu et al., 2014*; *Farnsworth et al., 2015*). Here we test whether columnar neuron subtypes arise from a young D$^+$ or old Ey$^+$ temporal window. As expected, all columnar neuron subtypes are labeled when the lexAop reporter is 'opened' in all INPs (*Figure 3A–D*). In contrast, when the lexAop reporter is 'opened' only in old INPs, we detect all 40 E-PG and all 30 PF-R adult neurons but no P-EN or P-FN neurons (*Figure 3E–H*). We conclude that all P-EN and P-FN neurons are born from young INP lineages, whereas all E-PG and PF-R neurons are born from old INP lineages (summarized in *Figure 3I*). Interestingly, the P-EN and P-FN columnar neurons have a highly similar developmental origin and project to similar CX neuropils (dendrites to PB, axons to Noduli; *Figure 3I*), whereas E-PG and PF-R columnar neurons have distinct developmental origins and share no similarities in neuropil targets, suggesting that developmental origin may be tightly linked to neuronal morphology and anatomical connectivity (see Discussion).

## The Eyeless temporal transcription factor promotes E-PG and PF-R molecular identity

Our birthdating results indicated that INP age might be a major determinant of CX columnar neuron morphology and connectivity. We next tested whether the TTF Eyeless, which is expressed by INPs

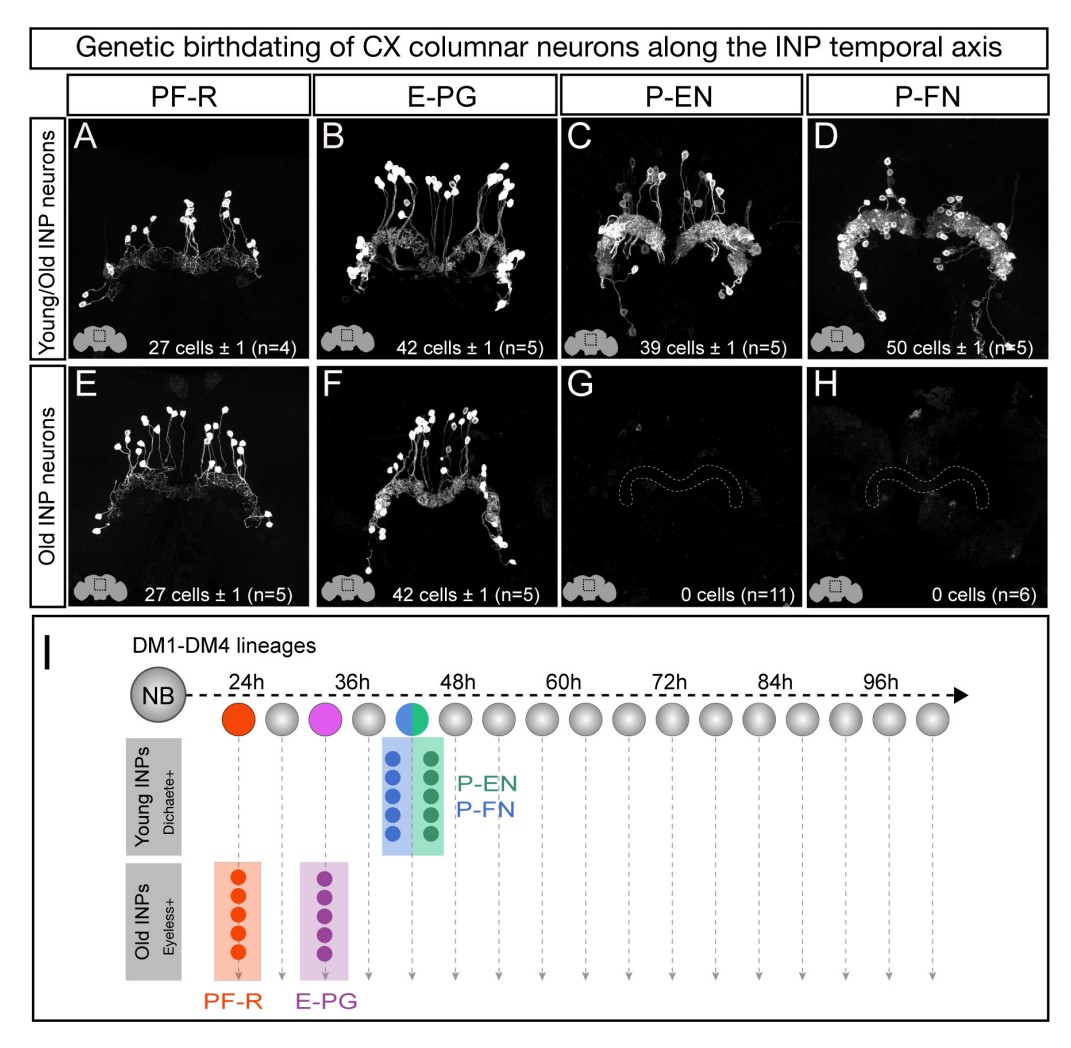

**Figure 3.** Each CX columnar neuron type arises exclusively from young or old INP lineages. (**A–D**) Columnar neuron cell bodies labeled by subtype-specific LexA lines derive from INP lineages (n = 5 for each experiment). Staining shows the volume containing cell bodies; thus most axon and dendrite projections are not visible. See *Figure 1—figure supplement 1A* for genetic details. (**E–H**) The PF-R and E-PG columnar neurons are generated by late INPs (n = 5 for each experiment), whereas the P-EN (n = 11) or P-FN (n = 6) neurons were not derived from old INPs and thus are fully derived from young INPs. Staining shows the volume containing cell bodies; thus most axon and dendrite projections are not visible. See *Figure 1—figure supplement 1C* for genetic details. (**I**) Summary of INP birthdating results.

DOI: https://doi.org/10.7554/eLife.43482.006

during the last half their lineage, specifies the identity of PF-R and E-PG neurons, which are born from Ey$^+$ INPs. To knock down Eyeless expression in INPs, we used an *eyeless* enhancer-Gal4 line (*R16B06-Gal4*) that is expressed in old INPs (*Farnsworth et al., 2015*) to drive a *UAS-Ey$^{RNAi}$* transgene that we previously showed eliminates all detectable Eyeless protein (*Bayraktar and Doe, 2013*).

In wild type adults, there are ~40 E-PG neurons and ~30 PF-R neurons (*Figure 4A,B*; quantified in G,H). In adults where Ey$^{RNAi}$ is expressed in old INPs, we found nearly complete loss of PF-R and E-PG neurons (*Figure 4D,E*; quantified in G,H); we suggest that these neurons are converted into an early-born INP progeny identity (for which we have no markers), but we can't rule out that they undergo apoptosis. In addition, we performed an antibody screen for neuronal markers of CX neuronal subtypes, and identified Toy as specifically marking all of the old INP-derived PF-R and E-PG neurons but none of the young INP-derived P-EN and P-FN neurons (*Figure 5—figure supplement 1*). Here we show that Toy$^+$ neurons generated by old INPs are also significantly reduced following

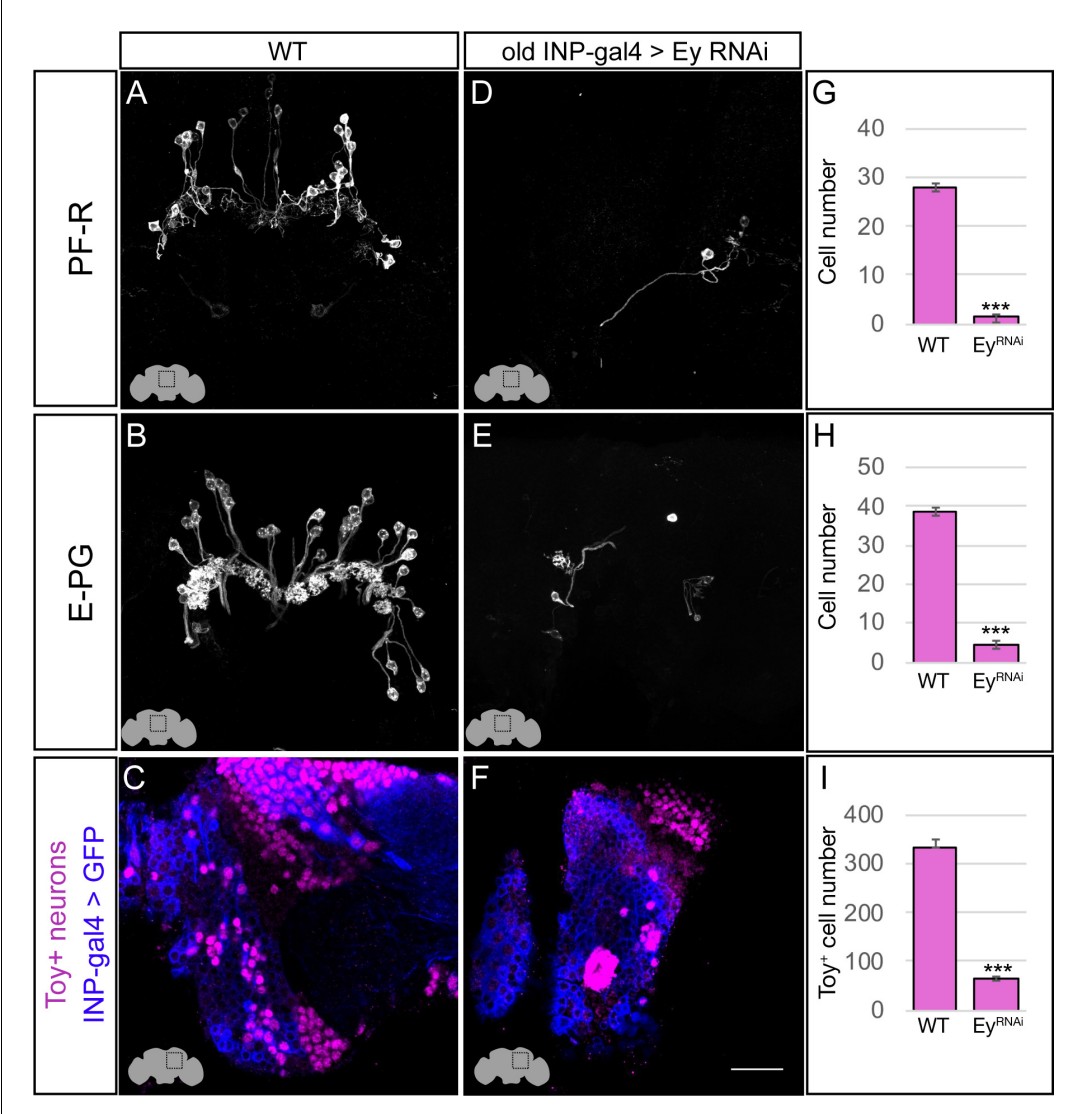

**Figure 4.** Eyeless promotes PF-R and E-PG molecular identity. (**A–C**) Wild-type PF-R, E-PG, and Toy[+] neurons in the dorsoposterior adult brain. PF-R and E-PG neurons detected by expression of neuron-specific LexA lines. See Materials and methods for genotypes. (**D–F**) Eyeless[RNAi] in INP lineages decreases the number of PF-R, E-PG, and Toy[+] late-born neurons in the dorsoposterior adult brain. See Materials and methods for genotypes. (**G–I**) Quantification (n = 5 for each experiment). ***, p<0.001. Scale bar, 20μm.

DOI: https://doi.org/10.7554/eLife.43482.007

Ey[RNAi] in old INPs (*Figure 4C,F*; quantified in I). We conclude that the Eyeless temporal transcription factor is required for the specification of PF-R and E-PG columnar neurons.

## The Eyeless temporal transcription factor represses P-EN and P-FN molecular identity

The P-EN and P-FN columnar neurons derive from early INP progeny, prior to the expression of Eyeless in later-born INPs, raising the question of whether Eyeless expression triggers a switch from early-born P-EN/P-FN production to late-born E-PG/PF-R production. To determine if Eyeless terminates production of early-born P-EN and P-FN columnar neurons, we expressed Ey[RNAi] in old INPs, and assayed for ectopic P-EN or P-FN neurons. In wild type adults, there are ~40 P-EN neurons and ~50 P-FN neurons (*Figure 5A,B*; quantified in G,H). In adults where Ey[RNAi] was expressed in old INPs, we found an over two-fold increase in the number of P-EN and P-FN neurons (*Figure 5D,E*; quantified in G,H). In addition, the antibody screen described above identified the transcription

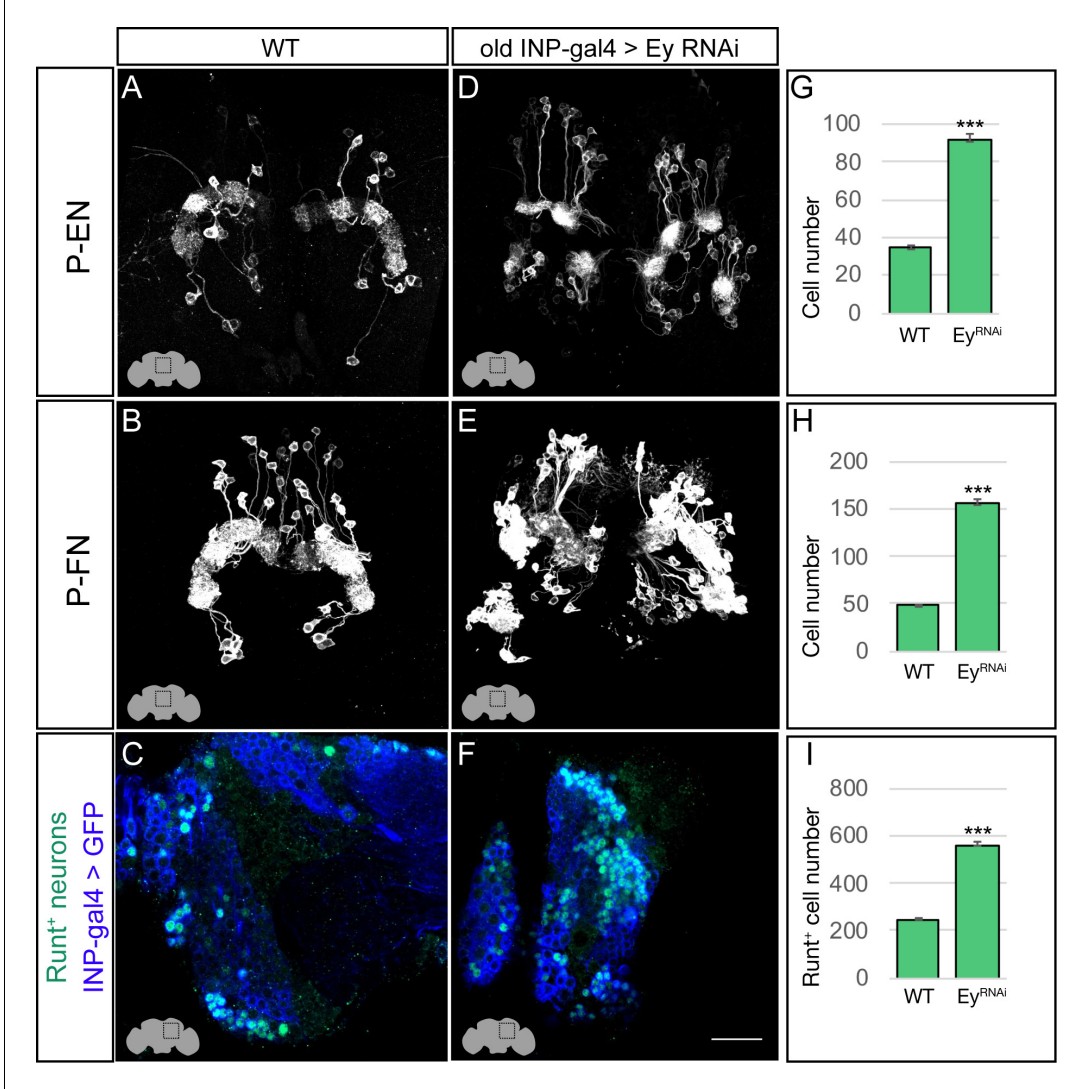

**Figure 5.** Eyeless represses P-EN and P-FN molecular identity. (**A–C**) Wild-type P-EN, P-FN, and Runt+ neurons in the dorsoposterior adult brain. P-EN and P-FN neurons detected by expression of neuron-specific LexA lines. See Materials and methods for genotypes. (**D–F**) Eyeless$^{RNAi}$ in INP lineages increases the number of P-EN, P-FN, and Runt+ late-born neurons in the dorsoposterior adult brain. See Materials and methods for genotypes. (**G–I**) Quantification (n = 5 for each experiment). ***, p<0.001. Scale bar, 20µm.

DOI: https://doi.org/10.7554/eLife.43482.008

The following figure supplement is available for figure 5:

**Figure supplement 1.** CX columnar neurons express either Runt or Toy in adult brain.

DOI: https://doi.org/10.7554/eLife.43482.009

factor Runt as specifically marking all early-born P-EN and P-FN neurons but none of the late-born E-PG and PF-R neurons (*Figure 5—figure supplement 1*). In wild type, there are ~220 Runt+ adult neurons made by INP progeny, but Ey$^{RNAi}$ led to a significant increase to ~580 Runt+ adult neurons (*Figure 5C,F*; quantified in I), consistent with a role for Eyeless in terminating production of young INP-derived neurons. We conclude that Eyeless maintains equal pools of E-PG and P-EN neurons by triggering a switch from early-born P-EN/P-FN neurons to late-born E-PG/PF-R neurons.

## Loss of Eyeless produces ectopic P-EN neurons with endogenous P-EN morphology and anatomical connectivity

Loss of Eyeless extends the production of P-EN neurons into an older stage of INP lineages, creating a mismatch between their molecular temporal identity (early) and their time of differentiation (late).

We tested whether the ectopic P-EN neurons have a neuronal morphology and anatomical connectivity characteristic of the endogenous early-born neurons, or whether their later birthdate results in different morphology or connectivity. We designed a genetic method for specifically labeling the ectopic late-born P-EN neurons – but not the endogenous early-born P-EN neurons – to trace their morphology and anatomical connectivity (*Figure 1—figure supplement 1D*).

As expected, control RNAi did not result in any ectopic P-EN neurons, although there were a few neurons labeled outside the central brain and a small pattern of fan-shaped body neurons (*Figure 6A–A'''*). In contrast, Eyeless[RNAi] specifically in old INP progeny resulted in the formation of sparse populations of 'late-born' ectopic P-EN neurons with projections into the PB, EB, and Noduli (*Figure 6B–B'''*). These are the same neuropils targeted by wild type early-born P-EN neurons. We conclude that ectopic late-born P-EN neurons have morphology indistinguishable from the normal early-born P-EN neurons (*Videos 1–2*).

To determine if the ectopic P-EN neurons have the same anatomical connectivity as the endogenous P-EN neurons, we expressed the pre-synaptic active zone marker Bruchpilot (Brp) specifically in the ectopic P-EN neurons. We found that ectopic P-EN neurons localized Brp to the EB and Noduli, but not to the PB. This is the same as in wild type P-EN neurons (*Figure 6C–C'*, summarized in *Figure 6E*). Furthermore, the ectopic P-EN neurons assembled into proper columns between glomeruli in the PB and tiles in the EB, precisely matching the morphology of endogenous P-EN neurons (*Figure 6D*; compare to Figure 8D1 in *Wolff et al., 2015*). Thus, ectopic P-EN neurons match the normal early-born P-EN neurons in molecular identity (R12D09-LexA[+]), morphology (PB, EB, Noduli projections), and anatomical connectivity (Brp puncta in EB and Noduli). Finally, we assayed the morphology of the ectopic P-FN neurons following Eyeless[RNAi]. We found that the expanded pool of P-FNs innervated the FB and NO, identical to endogenous P-FN neurons, resulting in an enlarged FB and NO (*Figure 6—figure supplement 1*). We conclude that reducing expression of the TTF Eyeless leads to a doubling of P-EN and P-FN neurons in the CX, which all have proper neuropil targeting. This shows that neuronal birth-date can be uncoupled from neuronal morphology, because we see P-EN and P-FN neurons born later than normal in the INP lineage, yet they establish morphology that mimics that of the endogenous, early-born P-EN and P-FN neurons (*Figure 6—figure supplement 2*).

## Transient Eyeless reduction impairs adult flight navigation behavior

Our finding that the temporal transcription factor Eyeless contributes to the development of CX columnar neurons raises the question of how Eyeless influences CX function. Recent work has shown that silencing adult E-PG neurons impairs flies' capacity to maintain an arbitrary heading to a bright spot resembling the sun (*Giraldo et al., 2018*; *Green et al., 2018*), a finding that we independently confirmed (*Figure 7—figure supplement 1A–D*). Based on these results, we hypothesized that Eyeless function during development may be required for adult E-PG function in sun navigation. To reduce Eyeless expression, we drove Eyeless[RNAi] in old INPs using *R16B06-Gal4*. Temporal control over Eyeless[RNAi] was achieved with the temperature-sensitive Gal4 inhibitor Gal80. We raised animals at the Gal80 permissive temperature (18°C) to prevent Eyeless[RNAi] expression and shifted to the non-permissive temperature (29°C) for 24 hr at the time E-PG neurons are born and differentiate (*Figure 7A*). Both control and Eyeless[RNAi] animals exposed to this regime had no major morphological defects in the central complex (EB shown in *Figure 7B,C*), indicating that E-PG neuron number is likely normal (see Discussion). We then examined how the transient reduction of Eyeless in larval INPs affected the ability of adult flies to maintain an arbitrary flight heading to a fictive sun (*Figure 7D*). We compared the sun headings of Eyeless[RNAi] flies that received the 29°C heat pulse with two control groups. One control group had an identical genotype but received no heat pulse (*Figure 7E*). A second control group received the heat pulse but Eyeless[RNAi] was replaced with mCherry[RNAi] (*Figure 7F*). In both control groups, we found that flies maintained arbitrary headings, as expected, with a slight bias towards headings where the sun was behind the fly (*Figure 7E,F,I*). In contrast, flies with transient Eyeless[RNAi] during E-PG development exhibited a marked frontal bias in their heading distribution, which was significantly more frontal than the control distributions (*Figure 7G,I*; p<0.01, permutation test). The control distributions were not significantly different from each other (p=0.49). Notably, although the heading distributions were distinct, the degree of stimulus stabilization – quantified by calculating the overall vector strength of each flight – was equivalent in the Eyeless[RNAi] genotype and controls (*Figure 7H*). Moreover, the Eyeless[RNAi]

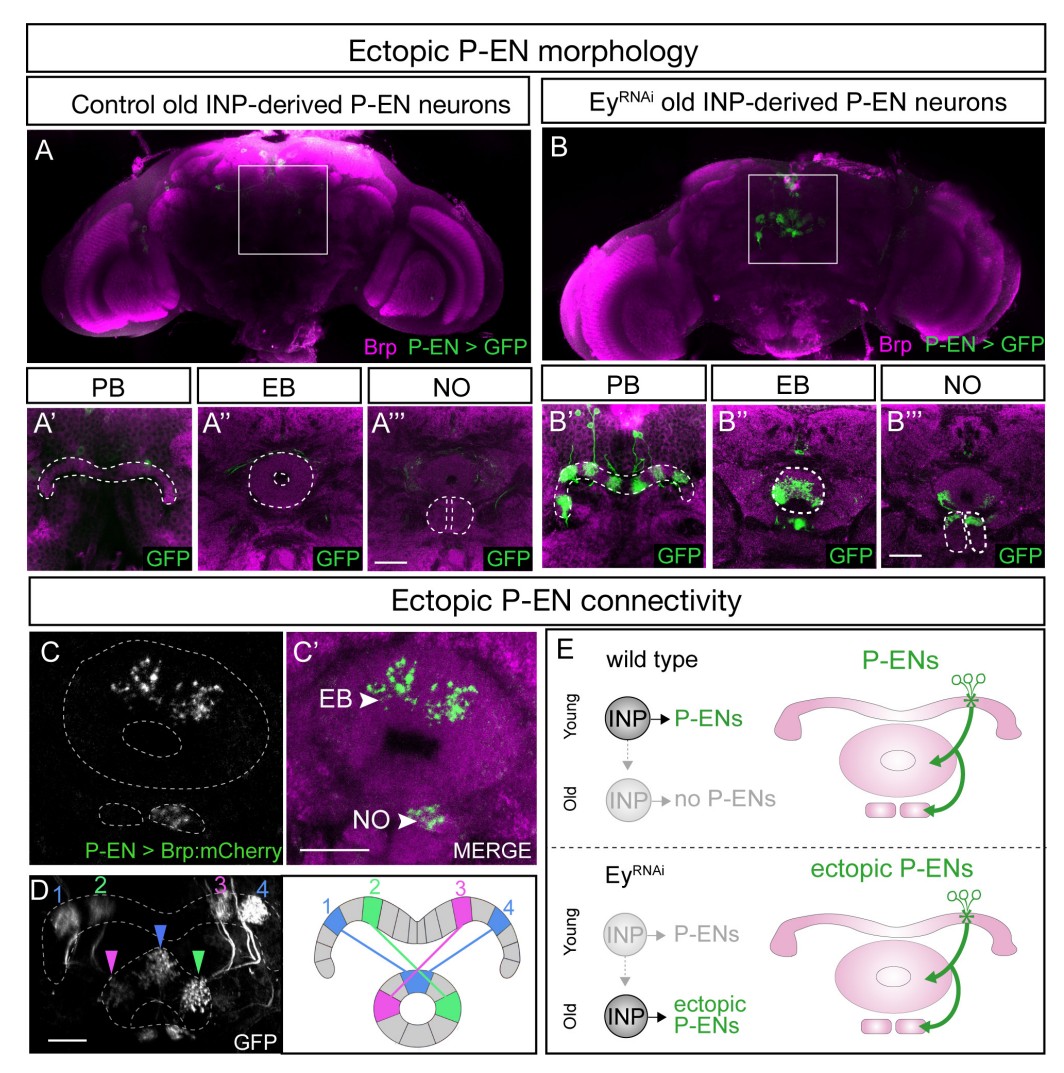

**Figure 6.** Eyeless[RNAi] produces late-born 'ectopic' P-EN neurons that have normal P-EN morphology and connectivity. (**A–A'''**) In wild-type adults, late INP clones do not label P-EN neurons in the adult brain (n = 5). See Materials and methods for details. PB, EB, and NO neuropils marked with dashed lines. Scale bars, 20μm (**A–B**). (**B–B'''**) In Eyeless[RNAi] adults, late INP clones produce ectopic late-born P-EN neurons, which project to the PB, EB, and Noduli (n = 5), similar to endogenous P-EN neurons (*Figure 1A,B*). PB, EB, and NO neuropils marked with dashed lines. (**C–C'**) In Eyeless[RNAi] adults, late INP clones produce ectopic late-born P-EN neurons, which localize the pre-synaptic marker Brp::mCherry to the EB and Noduli (n = 5), but not to the PB (not shown), similar to the endogenous P-EN neurons. Scale bars, 20μm (**C–D**). (**D**) Eyeless[RNAi] adult, showing stochastic labeling of four ectopic P-EN neurons (1-4) with normal PB and EB glomeruli targeting (compare to *Wolff and Rubin, 2018*). (**E**) Summary.
DOI: https://doi.org/10.7554/eLife.43482.010

The following figure supplements are available for figure 6:

**Figure supplement 1.** Eyeless[RNAi] produces late-born 'ectopic' P-FN neurons that have normal P-FN morphology.
DOI: https://doi.org/10.7554/eLife.43482.011

**Figure supplement 2.** Eyeless regulation of identity schematic.
DOI: https://doi.org/10.7554/eLife.43482.012

genotype and controls showed equivalent performance orienting to a dark vertical stripe (*Figure 7— figure supplement 1E–H*), similar to the effect of silencing adult E-PG neurons (*Giraldo et al., 2018*). This suggests that E-PG silencing and Eyeless[RNAi] induce similar, relatively specific navigation deficits rather than a more general deficiency in visual-motor flight control. Taken together, our results indicate that a transient loss of Eyeless specifically in old INPs causes specific deficits in adult

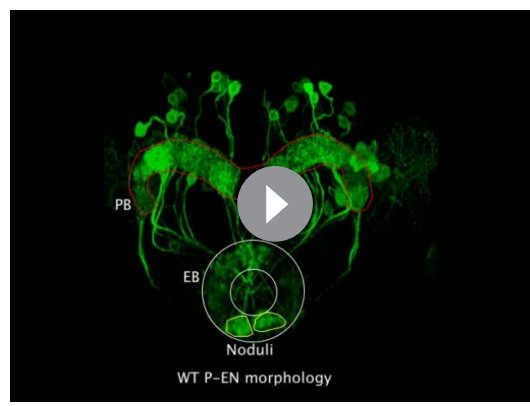

**Video 1.** Wild-type adult P-EN morphology wild-type young INP born P-EN neurons innervate distinct neuropil regions of the central complex. These include the PB, EB, and NO outlined in red, white, and yellow respectively.
DOI: https://doi.org/10.7554/eLife.43482.013

flight navigation to that of silencing E-PG neurons. Our findings therefore demonstrate the importance of Eyeless for CX function.

## The Eyeless target gene *toy* is required for E-PG axonal connectivity to the Gall

The TTF Eyeless is required to specify E-PG neuronal identity, but Eyeless does not persist in adult E-PG neurons, raising the question: What Eyeless target genes regulate E-PG connectivity and function? We focused on Twin of eyeless (Toy) which encodes a transcription factor whose expression is induced by Eyeless in old INPs (*Bayraktar and Doe, 2013*) and is maintained in their adult post-mitotic neuronal progeny. We used two previously characterized Gal4 drivers (*Kim et al., 2017*; *Lovick et al., 2017*) to express *UAS-toy^RNAi* specifically in post-mitotic E-PG neurons at different stages in development and confirmed that it removes all detectable Toy protein (*Figure 8* inset A, B).

We next determined if depleting Toy in post-mitotic larval E-PG neurons using *R19G02-Gal4 UAS-toy^RNAi* altered E-PG survival or morphology. Loss of Toy had no effect on E-PG neuronal number (n = 5, p=0.92) or on connectivity to the EB and PB (data not shown). In contrast, we observed greatly diminished E-PG axonal connectivity to the Gall, where in some cases the E-PG projections appeared nearly absent (n = 12, *Figure 8A–C*). We next removed Toy later, beginning ~24 hr after pupal formation using *ss00096-Gal4 UAS-toy^RNAi*, and observed no effect on E-PG neuronal number (n = 5, p=0.48) or projections to the EB, PB, or Gall (n = 6, *Figure 8D–F*). Surprisingly, however, loss of Toy produced a significant reduction in the levels of the pre-synaptic active zone marker Bruchpilot (Brp) in the Gall (*Figure 8G–I*). We conclude that Toy is required during larval stages for E-PG connectivity to the Gall, and is required in pupal stages for establishing or maintaining Brp levels at the E-PG axonal terminals in the Gall.

To determine how the loss of Toy during pupal stages affects CX function, we tested whether reduction of Toy in the E-PGs affected sun navigation. We observed no significant change in flies' heading distribution in relation to the sun stimulus, or in the degree to which they stabilized the sun stimulus (*Figure 8—figure supplement 1*). Therefore, the loss of Toy in pupal E-PG neurons and the associated reduction of Brp at E-PG axon terminals has no discernible effect on sun navigation.

## Discussion

### Developmental origin of CX columnar neurons

We have shown that distinct classes of CX columnar neurons have unique developmental origins within type II neuroblast lineages. We find that CX columnar neurons map to four bilateral type II neuroblast lineages (DM1-DM4), confirming previous work (*Wang et al., 2014*). Thus,

**Video 2.** Ey-RNAi ectopic adult P-EN morphology Ey-RNAi ectopic old INP born P-EN neurons innervate the same distinct neuropil regions of the central complex. These include the PB, EB, and NO outlined in red, white, and yellow respectively. Note, only even-numbered glomeruli are generated, indicating they are born after odd numbered glomeruli in the INP lineage, and extend when Eyeless is eliminated in old INPs.
DOI: https://doi.org/10.7554/eLife.43482.014

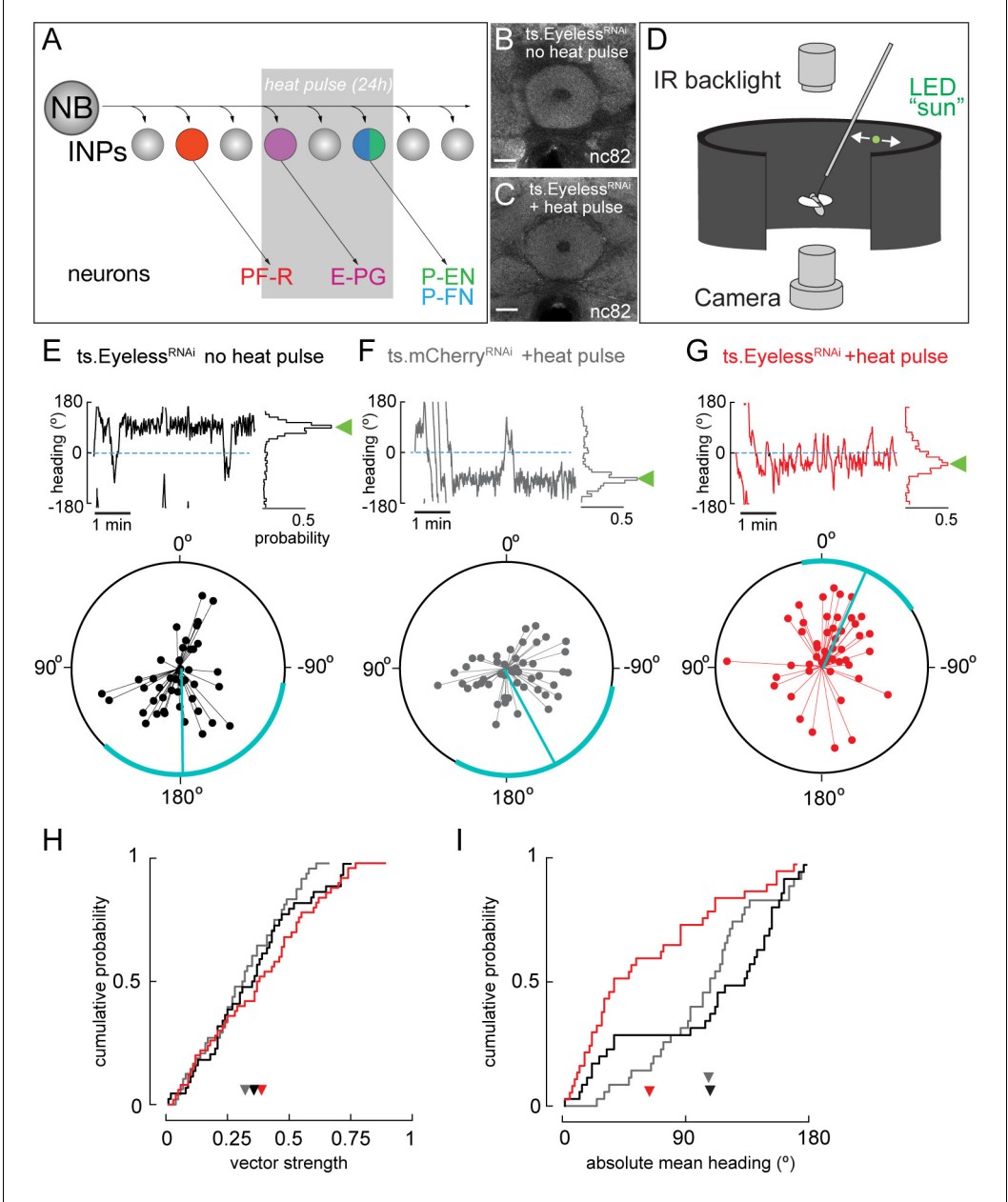

**Figure 7.** Transient loss of Eyeless during development impairs adult fly navigation. (**A**) Timing of Eyeless reduction in INP lineages. Transient inactivation of ts.Gal80 (29°C heat pulse; gray bar) results in transient Eyeless[RNAi] during the time in which E-PG neurons are normally generated. (**B,C**) The manipulation in A does not alter CX neuropil morphology as seen by nc82 (neuropil) staining (EB, shown; other neuropils, data not shown). (**D**) Schematic of experimental apparatus for sun navigation experiments. The wing stroke amplitudes of a tethered, flying fly were monitored with an IR camera; the stroke difference determined the angular velocity of a 2.4° sun stimulus. Modified from (***Giraldo et al., 2018***). (**E**) Example flight (top panel) and summary data (bottom panel) from ts.Eyeless[RNAi] control with no heat pulse. Top panel: left plot shows headings over 5 min flight; 0° is sun position in front of fly. Right histogram is distribution of headings in this example flight; sideways green triangle is the mean. Bottom panel: summary data, with each 5 min flight represented by radial lines. The angle of each line is the mean flight heading. The length of each line is vector strength of flight, varying from 0 (center of circle; no stimulus stabilization) to 1 (edge of circle; perfect stabilization). Each fly flew for two 5 min flights separated by a 5 min rest period. Cyan line shows mean heading, across flights with vector strength >0.2, as well as 95% confidence interval, calculated via resampling across flies. 44 flights, 22 flies total. (**F**) Example and summary data from ts. mcherry[RNAi] control. Same plotting convention as (**E**). 48 flights, 24 flies. (**G**) Example and summary data from ts.

*Figure 7 continued on next page*

*Figure 7 continued*

Eyeless[RNAi] flies with heat pulse. Same plotting convention as (**E,F**). 50 flights, 25 flies. (**H**) Cumulative probability distribution of vector strengths from both control groups (black and gray) and from experimental group (red). There was no significant difference between means (ts.Eyeless[RNAi] heat pulse, 0.34, red; ts.Eyeless[RNAi] no heat pulse, 0.31,black;ts.mcherry[RNAi] heat pulse 0.30,gray; p>0.1, permutation test). (**I**) Cumulative probability distribution of mean absolute headings. The heading distribution for the experimental group, ts.Eyeless[RNAi] heat pulse, was skewed significantly to frontal headings (mean 63.7°, 37 flights in 23 flies with vector strength >0.2) compared to control distributions (p<0.01, permutation test; ts.Eyeless[RNAi] no heat pulse, mean 107.9°, 35 flights in 19 flies; ts.mcherry[RNAi], mean 106.9°, 31 flights in 21 flies). Scale bars, 10µm.
DOI: https://doi.org/10.7554/eLife.43482.015
The following figure supplement is available for figure 7:

**Figure supplement 1.** Sun navigation is impaired when E-PG neurons are silenced; stripe fixation behavior is not altered by the loss of Eyeless.
DOI: https://doi.org/10.7554/eLife.43482.016

per brain there are eight parental neuroblasts that generate 30–50 neurons of each subtype, or 4–6 neurons per neuroblast. These 4–6 neurons could arise from 2 to 3 GMCs in a single INP lineage, or as 1 neuron from six different INPs; twinspot MARCM would be needed to determine their precise cell lineage. Our birth-dating results indicate that CX columnar neurons originate from distinct INPs born ~12 hr apart during larval life, except for P-EN and P-FN neurons whose similar birthdates suggest they may arise from the same INPs. Twin-spot MARCM analysis (*Lee and Luo, 1999*) would be necessary to determine whether P-EN and P-FN neurons arise from the same or different INPs. Interestingly, the two CX columnar neurons born at the same time (P-EN and P-FN) have axon projections intrinsic to the CX and target the same neuropils (PB and Noduli). In contrast, the two CX columnar neuron types born at different times (E-PG and P-FR) have axon projections extrinsic to the CX and target different neuropils (Gall and ROB). This raises the possibility that neuroblast temporal identity determines whether columnar neuron axon projections are intrinsic or extrinsic to the CX. More generally, the results suggest that neurons with similar temporal identity have matching connectivity.

We have mapped the birthdates of only four CX columnar neuron subtypes out of the 60 distinct neuronal subtypes innervating the CX (*Young and Armstrong, 2010*). Mapping these other neurons to their type II neuroblast and INP lineages is an important task for the future, which will help identify developmental correlates of neuronal morphology, connectivity, and function. Additionally, significant neuronal diversity may arise from GMCs dividing to make Notch[ON]/Notch[OFF] sibling neurons, which often have distinct morphology (*Truman et al., 2010*; *Lacin et al., 2014*; *Wang et al., 2014*; *Harris et al., 2015*). The role of Notch signaling in generating hemilineages within type II neuroblast progeny remains unexplored.

## Specification of CX columnar neurons

By mapping the developmental origins of four classes of columnar neurons innervating the central complex, we find that each class derives from a relatively tight window during the neuroblast lineage, and from either young or old INPs (*Figure 3I*). The fact that all of the four subtypes are restricted to early or late in the INP lineage suggests that the early/late lineage distinction is developmentally important, consistent with our finding that early/late INPs express different TTFs (Dichaete/Eyeless, respectively). Furthermore, mapping the lineage of each neuronal class allowed us to identify a correlation with developmental origin and neuronal morphology (neurons with similar birth-dates have similar morphology). Many other developmental windows have yet to be characterized, for example the neurons derived from young INPs prior to PF-R/E-PG production are unknown, and would be expected to be expanded in the absence of Eyeless; similarly, the neurons derived from the old INPs following production of the P-EN/P-FN neurons are unknown, and would be expected to be missing in the absence of Eyeless. We tested Dichaete and Grainy head for a role in specification of early INP-derived P-EN and P-FN neurons, but observed no phenotype (data not shown); this is unsurprising for Grainy head, because it is not expressed in the DM1 lineage (*Bayraktar and Doe, 2013*) which generates P-EN and P-FN neurons. In the future, our intersectional genetic approaches can be used to map the developmental origin of any neuronal subtype for which there exists an adult LexA driver line. For example, we have recently mapped the CX dorsal

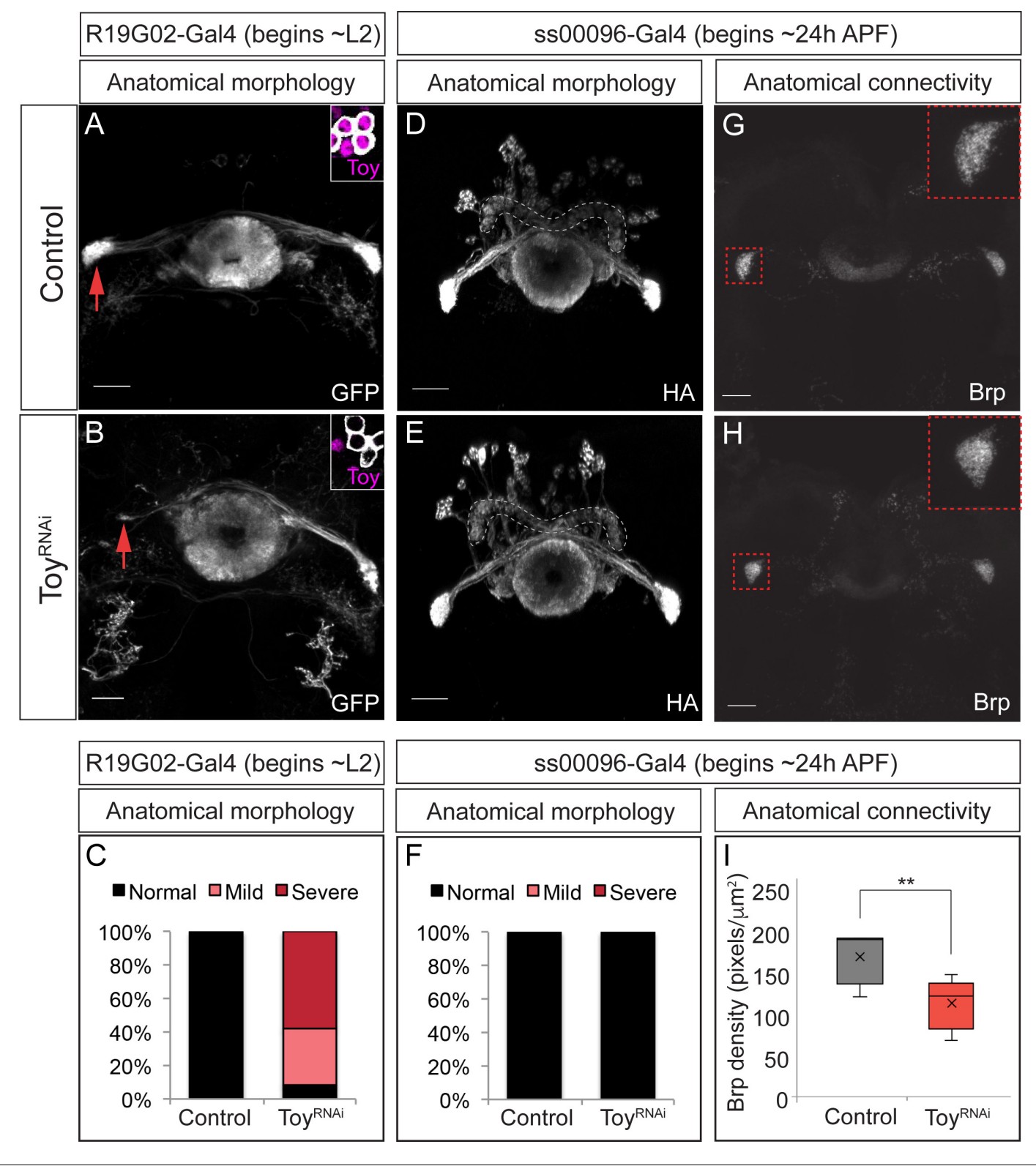

**Figure 8.** The Eyeless target gene Toy is required for E-PG axonal connectivity. (A–C) Loss of Toy in larvae reduces E-PG projections to the Gall in adults. (A) Wild type: *R19G02-Gal4* is first expressed at ~L2 and labels adult E-PG neurons; note projections to the EB (center) and Gall (left and right); PB, not shown (n = 6). Inset shows WT levels of Toy-protein expression. (B) *R19G02-Gal4 UAS-Toy^{RNAi}* reduces E-PG projections to the Gall (red arrow), yet projections to the EB and PB (not shown) remain intact (n = 12). Inset shows loss of Toy-protein expression. Quantification: mild, detectable

*Figure 8 continued on next page*

*Figure 8 continued*

reduction in the Gall in one hemisphere; severe, virtually complete loss of Gall. (D–F) Loss of Toy in pupae has no effect on E-PG projections. (D) In wild type, *ss00096 split-Gal4* is expressed ~24 hr after pupal formation and labels adult E-PG neurons; n = 5. (E) *ss00096 split-Gal4 UAS-Toy^{RNAi}* adults have normal projections to the EB (center), Gall (left and right), and PB (outlined); n = 5. (F) Quantification. (G–I) Loss of Toy in pupae reduces pre-synaptic levels of Brp in the Gall. (G,H) Genotypes as in D-E, showing that the pre-synaptic marker Brp is reduced in the E-PG axons targeting the Gall following ToyRNAi (n = 6). (I) Quantification. Scale bars, 20μm.

DOI: https://doi.org/10.7554/eLife.43482.017

The following figure supplement is available for figure 8:

**Figure supplement 1.** Reducing levels of Toy in E-PG neurons during pupal stages does not alter navigation to a fictive sun.

DOI: https://doi.org/10.7554/eLife.43482.018

fan-shaped body 'sleep neurons' (*Donlea et al., 2011*; *Ueno et al., 2012*; *Dubowy and Sehgal, 2017*; *Donlea et al., 2018*) to an old neuroblast developmental window (M. Syed, LS, and CQD, unpublished).

We have shown that Eyeless maintains a balance of early-born P-EN/P-FN neurons and late-born E-PG/PF-R neurons by triggering a switch from early-born to late-born neuronal identity. Loss of Eyeless generates fewer E-PG neurons and more P-EN neurons (*Figures 4* and *5*). We document the loss of late-born E-PGs here, but many other uncharacterized neurons are also likely to be lost, except during our heat pulse experiments where we tried to specifically target E-PG neurons (*Figure 7*). Similarly, we document the production of ectopic P-EN neurons in the absence of Eyeless, but many other early-born neuron populations are likely to be expanded. We considered performing clonal analysis to identify the neurons sharing an INP lineage with our four neural subtypes, but decided against it because INPs make morphologically different neurons at each division (*Wang et al., 2014*); we would not be able to map these neurons to early or late in the INP lineage, nor would we have molecular or genetic markers for these neurons. Determining the identity and birth-order of neurons within each INP lineage will be a difficult task for the future. Developing markers for the remainder of the 60+ different CX neuronal subtypes will be needed understand the breadth of Eyeless function in generating CX neuronal subtypes. Additional neuronal subtype markers will also be important to test the role of type II neuroblast candidate TTFs (*Ren et al., 2017*; *Syed et al., 2017*). We predict that at least some of these candidate TTFs will be required to specify the identity of the four columnar neuron classes described here.

We were interested in whether misexpression of Eyeless in young INPs was sufficient to induce ectopic late-born PF-R and E-PG neurons. We could not simply use *R9D11-Gal4* to misexpress Eyeless in young INPs, because we previously showed that in this genotype Ey translation is repressed in young INPs (*Farnsworth et al., 2015*). Thus, we permanently expressed Eyeless in INPs and their progeny (*R9D11-FLP, actin-FRT-stop-FRT-Gal4 UAS-eyeless*) but observed loss of all four neuronal subtypes (data not shown). Our interpretation is that permanent high level expression of Eyeless in INPs and their progeny leads to neuronal death, although we cannot rule out that Ey transforms all INP progeny into a late-born cell type that we lack markers to detect.

## Determinants of connectivity in CX columnar neurons

We have shown that the ectopic P-EN neurons formed due to reduced Eyeless levels have morphology and anatomical connectivity that matches the endogenous P-EN neurons (i.e. Brp^+ neurites to the EB and NO, and Brp^- neurites to the PB)(*Figure 6*). It is unknown, however, whether these ectopic P-ENs are functionally connected to the normal P-EN circuit partners. This could be resolved through functional imaging experiments testing whether ectopic P-ENs receive the innervation from E-PG or delta7 neurons like endogenous P-ENs (*Franconville et al., 2018*) or whether they form functional inputs to known E-PG downstream neurons (*Lin et al., 2013*; *Green et al., 2017*; *Turner-Evans et al., 2017*). Furthermore, we demonstrate that the Eyeless target gene Toy is required for E-PG axonal connectivity to the Gall. Future work could elucidate the target genes of Toy through RNA-seq that are required for assembling this connectivity, such as downstream cell surface molecules, thus linking INP temporal identity to a direct mechanism for neuronal connectivity in a highly conserved adult brain region.

## The effects of Eyeless manipulation on navigation behavior

We found that reducing Eyeless expression during early development (24–48 hr after larval hatching) causes a profound shift in how flies orient their flight relative to a fictive sun stimulus. Whereas control populations adopt a broad set of headings, with a slight bias for orientations where the sun is behind (*Figure 7E,F,I*), Eyeless[RNAi] flies choose flight directions where the sun is in front (*Figure 7G, I*). A similar shift to a more frontal heading distribution occurs when E-PG neurons are silenced, either following expression of the Kir2.1 inward rectifying channel (*Figure 7—figure supplement 1*; *Giraldo et al., 2018*) or with a synaptic transmission blocker in walking flies (*Green et al., 2018*). The consistent shift to a frontal heading after both E-PG silencing and Eyeless[RNAi] suggests that Eyeless[RNAi] affects navigation behavior via perturbation of E-PG neurons, although we cannot rule out an effect on unknown late-born neurons. Eyeless[RNAi] causes no gross deformities in the CX, suggesting E-PGs were not eliminated by Eyeless[RNAi] using this regime, as loss of all E-PG neurons produces severe EB defects (*Xie et al., 2017*). The developmental defects in E-PG neurons could be misexpression of ion channels or other functionally important molecules, rather than apoptosis. In contrast, genetic silencing likely affects all E-PG neurons (*Giraldo et al., 2018*). The fact that similar behavioral effects are induced by our more subtle Eyeless manipulation and E-PG silencing suggest that sun navigation is highly dependent on E-PG neuron activity. One difference between the behavioral effects of Eyeless[RNAi] and E-PG silencing is the degree to which flies stabilize the sun stimulus. Whereas silencing E-PG neurons significantly reduces the overall vector strength, a measure of the heading consistency within a flight (*Figure 7—figure supplement 1*; *Giraldo et al., 2018*; *Green et al., 2018*), there is no such reduction in vector strength in Eyeless[RNAi] flies (*Figure 7H*). This difference could be due to the more limited scope of the Eyeless manipulation or it could reflect some capacity of the adult CX to compensate for the larval developmental defect. Taken together, our findings demonstrate that a specific navigation behavior – arbitrary orientation to a sun stimulus – depends on the precise expression and function of the Eyeless TTF during larval development. These results raise the question of how other types of navigation depend on the development and function of CX neuronal subtypes.

## Materials and methods

### Key resources table

| Reagent type or resource | Designation | Source or reference | Identifiers | Additional information |
|---|---|---|---|---|
| Species (*Drosophila melanogaster*) | UAS-FLP | BDSC | #4539 | FLP enzyme under UAS control |
| Species (*Drosophila melanogaster*) | R9D11-Gal4 | BDSC | #40731 | Young INP Gal4 driver |
| Species (*Drosophila melanogaster*) | R37G12-lexA | BDSC | #52765 | PF-R lexA driver |
| Species (*Drosophila melanogaster*) | R60D05-lexA | BDSC | #52867 | E-PG lexA driver |
| Species (*Drosophila melanogaster*) | R12D09-lexA | BDSC | #54419 | P-EN lexA driver |
| Species (*Drosophila melanogaster*) | R16D01-lexA | BDSC | #52503 | P-FN lexA driver |
| Species (*Drosophila melanogaster*) | lexAop(FRT.stop)mCD8::GFP | BDSC | #57588 | FLP-out membrane bound GFP under lexAop control |
| Species (*Drosophila melanogaster*) | ts.Tubulin-Gal80 (20) | BDSC | #7019 | temperature sensitive Gal80 |
| Species (*Drosophila melanogaster*) | 20xUAS-FLP.PEST | BDSC | #55807 | FLP enzyme under 20xUAS control |
| Species (*Drosophila melanogaster*) | OK107-Gal4 | BDSC | #854 | Eyeless enhancer trap Gal4 for old INPs |

*Continued on next page*

*Continued*

| Reagent type or resource | Designation | Source or reference | Identifiers | Additional information |
|---|---|---|---|---|
| Species (*Drosophila melanogaster*) | UAS-mCherryRNAi | BDSC | #35787 | Control RNAi under UAS control |
| Species (*Drosophila melanogaster*) | UAS-Eyeless[RNAi] | BDSC | #32486 | Eyeless RNAi under UAS control |
| Species (*Drosophila melanogaster*) | 13xlexAop-myr::GFP | BDSC | #32210 | membrane bound GFP under 13xlexAop control |
| Species (*Drosophila melanogaster*) | R16B06-Gal4 | BDSC | #45811 | old INP Gal4 driver |
| Species (*Drosophila melanogaster*) | UAS-FLP, Act5c(FRT.CD2)Gal4; ; R12E09-Gal4, UAS-mCD8::GFP | This work | | INP immortalization driver expressing membrane bound GFP |
| Species (*Drosophila melanogaster*) | lexAop(FRT.stop)HA:: CD4.T2A.Brp.mCherry | BDSC | #56518 | FLP-out fluorensent labeling of Brp |
| Species (*Drosophila melanogaster*) | ss00096-Gal4 | Rubin Lab (Janelia) | | E-PG split Gal4 driver |
| Species (*Drosophila melanogaster*) | Empty-vector split Gal4 | Rubin Lab (Janelia) | | Control split Gal4 driver |
| Species (*Drosophila melanogaster*) | UAS-Toy[RNAi] | BDSC | #33679 | Toy RNAi under UAS control |
| Species (*Drosophila melanogaster*) | lexAop.tdTomato.myr, brp(FRT.stop)V5-2A-lexA-VP16 | BDSC | #56142 | STaR FLP-out labeling of synaptic terminals |
| Species (*Drosophila melanogaster*) | 10xUAS-myr::HA | BDSC | #62145 | membrane bound HA under UAS control |
| Species (*Drosophila melanogaster*) | R19G02-Gal4 | BDSC | #48860 | developmental E-PG Gal4 |
| Species (*Drosophila melanogaster*) | UAS-Kir2.1 | *Giraldo et al., 2018* | | Inward rectifying K + channel under UAS control |
| Antibody, polyclonal | Chicken anti-GFP | Abcam (Eugene, OR) | | 1:1000 |
| Antibody, polyclonal | Rabbit anti-Toy | Desplan lab (NYU) | | 1:1000 |
| Antibody, polyclonal | Guinea-pig anti-Runt | Desplan lab (NYU) | | 1:1000 |
| Antibody, monoclonal | Mouse anti-nc82 | DSHB (Iowa City, IA) | | 1:50 |
| Antibody, polyclonal | Rabbit anti-V5 | Cell Signaling (Danvers MA) | | 1:400 |
| Antibody, polyclonal | Rabbit Anti HA | Columbia Biosciences (Frederick MD) | | 1:400 |
| Antibody, polyclonal | Secondary antibodies | Thermofisher (Eugene, OR) | | 1:400 |

| Fly genotypes used in each experiment | Figure | Synopsis |
|---|---|---|
| UAS-FLP; R9D11-Gal4 X R37G12-lexA; lexAop(FRT.stop)mCD8::GFP | *Figure 1D* | PF-R labeling |
| UAS-FLP; R9D11-Gal4 X R60D05-lexA; lexAop(FRT.stop)mCD8::GFP | *Figure 1D* | E-PG labeling |
| UAS-FLP; R9D11-Gal4 X R12D09-lexA; lexAop(FRT.stop)mCD8::GFP | *Figure 1D* | P-EN labeling |
| UAS-FLP; R9D11-Gal4 X R16D01-lexA; lexAop(FRT.stop)mCD8::GFP | *Figure 1D* | P-FN labeling |
| 20XUAS-FLP.PEST; ts.Tubulin-Gal80 (20); R9D11-Gal4 X R37G12-lexA; lexAop(FRT.stop)mCD8::GFP | *Figure 2A* | PF-R birthdating |

*Continued*

| Fly genotypes used in each experiment | Figure | Synopsis |
|---|---|---|
| 20XUAS-FLP.PEST; ts.Tubulin-Gal80 (20); R9D11-Gal4 X R60D05-lexA; lexAop(FRT.stop)mCD8::GFP | *Figure 2A* | E-PG birthdating |
| 20XUAS-FLP.PEST; ts.Tubulin-Gal80 (20); R9D11-Gal4 X R12D09-lexA; lexAop(FRT.stop)mCD8::GFP | *Figure 2A* | P-EN birthdating |
| 20XUAS-FLP.PEST; ts.Tubulin-Gal80 (20); R9D11-Gal4 X R16D01-lexA; lexAop(FRT.stop)mCD8::GFP | *Figure 2A* | P-FN birthdating |
| UAS-FLP; R9D11-Gal4 X R37G12-lexA; lexAop(FRT.stop)mCD8::GFP | *Figure 3A* | PF-R labeling |
| UAS-FLP; R9D11-Gal4 X R60D05-lexA; lexAop(FRT.stop)mCD8::GFP | *Figure 3B* | E-PG labeling |
| UAS-FLP; R9D11-Gal4 X R12D09-lexA; lexAop(FRT.stop)mCD8::GFP | *Figure 3C* | P-EN labeling |
| UAS-FLP; R9D11-Gal4 X R16D01-lexA; lexAop(FRT.stop)mCD8::GFP | *Figure 3D* | P-FN labeling |
| OK107-Gal4 × 20 XUAS-FLP.PEST; R37G12-lexA; lexAop(FRT.stop)mCD8::GFP | *Figure 3E* | PF-R labeling old INP |
| OK107-Gal4 × 20 XUAS-FLP.PEST; R60D05-lexA; lexAop(FRT.stop)mCD8::GFP | *Figure 3F* | E-PG labeling old INP |
| OK107-Gal4 × 20 XUAS-FLP.PEST; R12D09-lexA; lexAop(FRT.stop)mCD8::GFP | *Figure 3G* | P-EN labeling old INP |
| OK107-Gal4 × 20 XUAS-FLP.PEST; R16D01-lexA; lexAop(FRT.stop)mCD8::GFP | *Figure 3H* | P-FN labeling old INP |
| 13xlexAop-myr::GFP; UAS-mCherry[RNAi] X R37G12-lexA; R16B06-Gal4 | *Figure 4A* | PF-R labeling WT |
| 13xlexAop-myr::GFP; UAS-mCherry[RNAi] X R60D05-lexA; R16B06-Gal4 | *Figure 4B* | E-PG labeling WT |
| UAS-FLP, Act5c(FRT.CD2)Gal4; ; R12E09-Gal4, UAS-mCD8::GFP X UAS-mCherry[RNAi] | *Figure 4C* | INP lineage tracing WT |
| 13xlexAop-myr::GFP; UAS-Eyeless[RNAi] X R37G12-lexA; R16B06-Gal4 | *Figure 4D* | PF-R labeling Ey-RNAi |
| 13xlexAop-myr::GFP; UAS-Eyeless[RNAi] X R60D05-lexA; R16B06-Gal4 | *Figure 4E* | E-PG labeling Ey-RNAi |
| UAS-FLP, Act5c(FRT.CD2)Gal4; ; R12E09-Gal4, UAS-mCD8::GFP X UAS-Eyeless[RNAi] | *Figure 4F* | INP lineage tracing Ey-RNAi |
| 13xlexAop-myr::GFP; UAS-mCherry RNAi X R12D09-lexA; R16B06-Gal4 | *Figure 5A* | P-EN labeling WT |
| 13xlexAop-myr::GFP; UAS-mCherry RNAi X R16D01-lexA; R16B06-Gal4 | *Figure 5B* | P-FN labeling WT |
| UAS-FLP, Act5c(FRT.CD2)Gal4; ; R12E09-Gal4, UAS-mCD8::GFP X UAS-mCherry[RNAi] | *Figure 5C* | INP lineage tracing WT |
| 13xlexAop-myr::GFP; UAS-Eyeless[RNAi] X R12D09-lexA; R16B06-Gal4 | *Figure 5D* | P-EN labeling Ey-RNAi |
| 13xlexAop-myr::GFP; UAS-Eyeless[RNAi] X R16D01-lexA; R16B06-Gal4 | *Figure 5E* | P-FN labeling Ey-RNAi |
| UAS-FLP, Act5c(FRT.CD2)Gal4; ; R12E09-Gal4, UAS-mCD8::GFP X UAS-Eyeless[RNAi] | *Figure 5F* | INP lineage tracing Ey-RNAi |

*Continued on next page*

*Continued*

| Fly genotypes used in each experiment | Figure | Synopsis |
| --- | --- | --- |
| R37G12-lexA X 13xlexAop-myr::GFP | *Figure 5A–A'''*, *Figure 5—figure supplement 1* | PF-R labeling |
| R60D05-lexA X 13xlexAop-myr::GFP | *Figure 5B–B'''*, *Figure 5—figure supplement 1* | E-PG labeling |
| R12D09-lexA X 13xlexAop-myr::GFP | *Figure 5C–C'''*, *Figure 5—figure supplement 1* | P-EN labeling |
| R16D01-lexA X 13xlexAop-myr::GFP | *Figure 5D–D'''*, *Figure 5—figure supplement 1* | P-FN labeling |
| R12D09-lexA; lexAop(FRT.stop)mCD8::GFP; OK107-Gal4 × 20 XUAS-FLP.PEST; ts.Tubulin-Gal80 (20); UAS-mCherry$^{RNAi}$ | *Figure 6A–A'''* | Ectopic P-EN WT |
| R12D09-lexA; lexAop(FRT.stop)mCD8::GFP; OK107-Gal4 × 20 XUAS-FLP.PEST; ts.Tubulin-Gal80 (20); UAS-Eyeless$^{RNAi}$ | *Figure 6B–B'''* | Ectopic P-EN Ey-RNAi |
| R12D09-lexA; lexAop(FRT.stop)HA::CD4.T2A.Brp.mCherry; OK107-Gal4 × 20 XUAS-FLP.PEST; ts.Tubulin-Gal80 (20); UAS-Eyeless$^{RNAi}$ | *Figure 6C* | Ectopic P-EN brp Ey-RNAi |
| R16D01-lexA; R16B06-Gal4 × 13xlexAop-myr::GFP; UAS-mCherry$^{RNAi}$ | *Figure 6A*, *Figure 6—figure supplement 1* | WT P-FN neuron morph. |
| R16D01-lexA; R16B06-Gal4 × 13xlexAop-myr::GFP; UAS-Eyeless$^{RNAi}$ | *Figure 6B*, *Figure 6—figure supplement 1* | Ey-RNAi P-FN neuron morph. |
| R16B06-Gal4 X ts.Tubulin-Gal80 (10); UAS-Eyeless$^{RNAi}$ | *Figure 7B* | no heat pulse control nc82 |
| R16B06-Gal4 X ts.Tubulin-Gal80 (10); UAS-Eyeless$^{RNAi}$ | *Figure 7C* | heat pulse nc82 |
| R16B06-Gal4 X ts.Tubulin-Gal80 (10); UAS-Eyeless$^{RNAi}$ | *Figure 7E* | no heat pulse control behavior |
| R16B06-Gal4 X ts.Tubulin-Gal80 (10); UAS-mCherry$^{RNAi}$ | *Figure 7F* | heashock control behavior |
| R16B06-Gal4 X ts.Tubulin-Gal80 (10); UAS-Eyeless$^{RNAi}$ | *Figure 7G* | heat pulse exp. behavior |
| 10xUAS-myr::GFP; R19G02-Gal4 X UAS-mCherry$^{RNAi}$ | *Figure 8A* | E-PG dev. driver control |
| 10xUAS-myr::GFP; R19G02-Gal4 X UAS-Toy$^{RNAi}$ | *Figure 8B* | E-PG dev. driver Toy-LOF |
| 10xUAS-myr::HA; ss00096-Gal4 X UAS-mCherry$^{RNAi}$ | *Figure 8D* | E-PG split driver control |
| 10xUAS-myr::HA; ss00096-Gal4 X UAS-Toy$^{RNAi}$ | *Figure 8E* | E-PG split driver Toy-LOF |
| 20XUAS-FLP.PEST; ss00096-Gal4 X lexAop-tdTomato.myr, brp(FRT.stop)V5-2A-lexA-VP16; UAS-mCherry$^{RNAi}$ | *Figure 8G* | E-PG STaR control |
| 20XUAS-FLP.PEST; ss00096-Gal4 X lexAop-tdTomato.myr, brp(FRT.stop)V5-2A-lexA-VP16; UAS-Toy$^{RNAi}$ | *Figure 8H* | E-PG STaR Toy-LOF |
| R16B06-Gal4 X ts.Tubulin-Gal80 (10); UAS-Eyeless$^{RNAi}$ | *Figure 7E*, *Figure 7—figure supplement 1* | no heat pulse control behavior |

*Continued on next page*

*Continued*

| Fly genotypes used in each experiment | Figure | Synopsis |
| --- | --- | --- |
| R16B06-Gal4 X ts.Tubulin-Gal80 (10); UAS-mCherry[RNAi] | *Figure 7F*, *Figure 7—figure supplement 1* | heat pulse control behavior |
| R16B06-Gal4 X ts.Tubulin-Gal80 (10); UAS-Eyeless[RNAi] | *Figure 7G*, *Figure 7—figure supplement 1* | heat pulse exp. behavior |
| Empty Split-Gal4 x UAS-Kir2.1 | *Figure 7—figure supplement 1* | E-PG control behavior |
| ss00096 Split-Gal4 x UAS-Kir2.1 | *Figure 7—figure supplement 1* | E-PG silenced behavior |
| Empty Split-Gal4 x UAS Toy[RNAi] | *Figure 8—figure supplement 1* | Toy-RNAi control behavior |
| ss00096 Split-Gal4 x UAS Toy[RNAi] | *Figure 8—figure supplement 1* | Toy-RNAi exper. behavior |

## Standardizing larval development at different temperatures

All larvae were grown at 25°C unless noted, and all hours after larval hatching are standardized to grow wild type at 25°C based on published conversions: 18°C is 2.25x slower than 25°C, and 29°C is 1.03x faster than 25°C (*Powsner, 1935*).

## Immunohistochemistry

Primary and secondary antibodies, see Key Resources Table, above. Adult brain dissections were conducted at room temperature with 2–5 day old adult females. Adult brains were dissected in 2% formaldehyde solution in Phosphate-Buffered Saline with. 5% Triton-X (PBST) and incubated for 55 min before applying an overnight block solution (5% Goat/Donkey serum, Vector Laboratories) at 4°C. Brains were then washed in PBST for one hour before applying an overnight primary mix at 4°C. Then, brains were washed for one hour at room temperature in PBST, before applying an overnight secondary mix at 4°C. Finally, brains were mounted in 90% glycerol, and imaged immediately.

## Imaging, data acquisition, and image analysis

Fluorescent images were acquired on a Zeiss LSM 700. Adult brain cell counting was performed using the Fiji cell counter plug in, and statistical analysis (Student's T test) was done in Excel. Figures were assembled in Illustrator (Adobe). Relative Brp-density was quantified in Fiji; maximum intensity projections were made, a rectangular ROI selected around the Gall, and a histogram plot of pixel intensity was generated. Background for image was calculated in neighboring ROIs and subtracted from each individual histogram plot-value. Intensity values were then summed together to calculate total intensity, and this was divided by Gall total area, calculated manually in Fiji using polygon selection tool. Qualitative measurements of Gall defects were made by observing whether the total area of the Gall had been reduced, or entirely eliminated, through visual observations in FIJI.

## Fly tethering for flight behavior

We used 3–4 day old females for behavioral experiments. We tethered flies under cold anesthesia, gluing a tungsten wire to the anterior notum with UV-cured glue (Bondic). The head was immobilized relative to the body with a small amount of glue between the head and thorax. Flies recovered for at least 20 min prior to behavioral testing.

## Flight arena and behavioral protocol

We coupled the angular velocity of a visual stimulus that was presented via LED panels to the continuously measured difference in wing stroke amplitude. Stroke amplitude was tracked at 60 Hz via Kinefly, a previously described video tracking system (*Suver et al., 2016*). A digital camera equipped with macro lens (Computar MLM3x-MP) and IR filter (Hoya) captured wing images from a 45° mirror positioned beneath the fly. Backlit illumination of wings was provided by a collimated infrared LED

above fly (Thorlabs #M850L3). We displayed visual stimuli using a circular arena of 2 rows of 12 LED panels (24 panels total). Each panel had 64 pixels (Betlux #BL-M12A881PG-11, λ = 525 nm) and was controlled using hardware and firmware (IORodeo.com) as previously described (*Giraldo et al., 2018*).The gain between stimulus angular velocity and wing stroke amplitude difference was 4.75°/s per degree of wing stroke difference. The sun stimulus was a single LED pixel which is ~2.4° on fly retina (*Giraldo et al., 2018*), ~30 deg above fly. The stripe was four pixels wide and 16 pixels high (15° by 60°). Flight experiments were controlled in the ROS environment. Incoming video was collected at 60 Hz and stimulus position data (i.e. the flight heading) at 200 Hz. In each experiment, flies navigated in closed loop to the sun stimulus in two distinct 5 min trials, which were separated by a 5 min rest period, during which we gave flies a small piece of paper to manipulate with their legs. Following the second sun flight, flies flew for 5 min in closed loop to the stripe stimulus. We discarded flights in which a fly stopped flying more than once during a sun or stripe presentation; furthermore, we discarded flights from flies that did not complete the two 5 min sun flights.

### Behavioral data analysis

All data analysis was conducted using custom scripts in Python, (*Warren, 2019*; archived at https://github.com/elifesciences-publications/elife_2019). The circular mean heading of a flight was computed as the angle of resultant vector obtained via vector summation, treating each angular heading measurement as a unit vector. To determine the vector strength, we normalized the length of the resultant vector by the number of individual headings.

### Statistics

Data represent mean ± standard deviation. Two-tailed Student's t-tests were used to assess statistical significance of anatomical data, with *p<0.05; **p<0.01; ***p<0.001. To determine the significance of differences in the mean of the vector strength and heading distribution between groups, we used Fisher's exact test with 10,000 permutations (*Fisher, 1937*). To avoid pseudoreplication, we permuted across flies rather than flights. We computed a 95% confidence interval of the circular mean of each heading distribution by bootstrapping from the observed data. For each experimental condition, we resampled with replacement from the observed flight data (resampling across flies not flights) to create 10,000 distributions of matched size to the observed data set. Confidence intervals were computed from the circular means of these 10,000 distributions. For analysis of the heading distributions and confidence intervals, we considered flights with a vector strength above a minimum threshold of 0.2.

## Acknowledgements

We thank the Tanya Wolff and Gerry Rubin for fly stocks. We thank Michael Dickinson for support in conducting the behavioral experiments. We thank Mubarak Syed, Emily Sales, Tanya Wolff, Claude Desplan, John Tuthill, and Michael Dickinson for comments on the manuscript. We thank Cooper Doe for assistance with brain dissections. Stocks obtained from the Bloomington Drosophila Stock Center (NIH P40OD018537) were used in this study. Funding was provided by HHMI (CQD, LS, TLW), NIH R37HD27056 (LS), and NIH T32HD007348 (LS).

## Additional information

### Funding

| Funder | Grant reference number | Author |
| --- | --- | --- |
| Howard Hughes Medical Institute | | Luis F Sullivan |
| National Institutes of Health | T32HD007348 | Luis F Sullivan |
| National Institutes of Health | R37HD27056 | Luis F Sullivan |

The funders had no role in study design, data collection and interpretation, or the decision to submit the work for publication.

## Author contributions
Luis F Sullivan, Conceptualization, Resources, Formal analysis, Validation, Investigation, Methodology, Writing—original draft, Writing—review and editing; Timothy L Warren, Conceptualization, Formal analysis, Validation, Investigation, Methodology, Writing—original draft, Writing—review and editing; Chris Q Doe, Conceptualization, Data curation, Supervision, Funding acquisition, Writing—original draft, Project administration, Writing—review and editing

## Author ORCIDs
Luis F Sullivan (D) http://orcid.org/0000-0003-0149-0999
Timothy L Warren (D) http://orcid.org/0000-0002-4429-4106
Chris Q Doe (D) http://orcid.org/0000-0001-5980-8029

## Decision letter and Author response
Decision letter https://doi.org/10.7554/eLife.43482.023
Author response https://doi.org/10.7554/eLife.43482.024

# Additional files

## Supplementary files
• Transparent reporting form
DOI: https://doi.org/10.7554/eLife.43482.019

## Data availability
All imaging data generated or analyzed during this study are included in the manuscript and supporting files. The flight behavioral data is archived in a Dryad repository: doi:10.5061/dryad.45177sc. Code is available at https://github.com/timothylwarren/elife_2019 (copy archived at https://github.com/elifesciences-publications/elife_2019).

The following dataset was generated:

| Author(s) | Year | Dataset title | Dataset URL | Database and Identifier |
| --- | --- | --- | --- | --- |
| Luis F Sullivan, Timothy L Warren, Chris Q Doe | 2019 | Data from: Temporal identity establishes columnar neuron morphology, connectivity, and function in a Drosophila navigation circuit | https://doi.org/10.5061/dryad.45177sc | Dryad, 10.5061/dryad.45177sc |

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
