## [Decision Letter]

Thank you for submitting your article "Temporal identity establishes columnar neuron morphology, connectivity, and function in a *Drosophila* navigation circuit" for consideration by *eLife*. Your article has been reviewed by two peer reviewers, and the evaluation has been overseen by K VijayRaghavan as the Reviewing and Senior Editor. The reviewers have opted to remain anonymous.

The reviewers have discussed the reviews with one another and the Reviewing Editor has drafted this decision to help you prepare a revised submission.

Summary:

In this manuscript, Sullivan et al. map the developmental origin of four central complex (CX) columnar neuron types from Type II neuroblast lineages in *Drosophila*. They first devise a genetic method to make use of pupal/adult CX neuron type markers to confirm that columnar neuron types are derived from DM1-4 Type II neuroblasts. They narrow down the time window when these neuron types are produced from neuroblasts that express Imp, a marker of young neuroblasts. Since Type II neuroblasts produce intermediate neural progenitors (INPs) that proceed through a traditional temporal transcription factor series, the authors show that P-EN and P-FN are born from young INPs (Dichaete window) while E-PG and PF-R are born from old INPs (Eyeless window). The authors then proceed to show that loss of Ey via RNAi leads to additional early born neurons and fewer late born neurons. Importantly, the additional early born neurons appear to have normal morphology and anatomical connectivity. The authors link the specification of late born neurons to an adult flight behavior by showing that loss of Ey via RNAi in E-PG neurons, previously shown to be important for flight heading, leads to similar flight defects as silencing E-PG neurons. Finally, since Ey is not maintained in differentiating neurons, the authors show that loss of Toy, a downstream target of Ey, leads to connectivity defects to the Gall (a synaptic target) by E-PG neurons and to similar navigation behavioral defects.

In all, this is a well-written manuscript that begins to describe the developmental origin of neuron types with a role in interesting flight behavior. However, as it stands, the paper (while a solid and interesting contribution) needs some work to go beyond a description of the early/late, neuroblast or INP origin of these four neuron types. It is difficult to figure out how one, or a few of these four Type II neuroblasts do generate so many of all these four neurons. Therefore, there are several questions that must be answered to increase the impact of this paper as there are important points that are missing that should be addressed by the authors prior to publication. There are also a few errors (mostly labeling) that need correcting. To provide broad new insights, this paper must describe the strategy used by nature to produce these four related neurons. We feel the essential changes can be completed in a reasonable time but if this presents a problem, please let us know.

Essential revisions:

1) The tracing of most if not all CX columnar neurons to TypeII NB lineages has already been shown by the Lee and Reichert labs. To go one step further, the authors must trace specific CX neuron types (PF-R, E-PG, P-EN, P-FN) to a specific Type II neuroblast. Since many neurons (30-50) of each type are produced, which neuroblasts produce them and how do they do it? Does each neuroblast produce an early INP and a late INP neuron type? But then, how is it possible to produce 50 neurons? Do GMCs produce two identical neurons, or are they distinct by Notch activity? Does this mean that each neuroblast produces identical INPs for multiple early time windows? Alternatively, as in the case of the mushroom body, all four neuroblasts might act identically and each could produce the same early INPs a dozen times, and each INP would produce four neurons (two D early neurons, maybe as two hemi-lineages, and two Ey late neurons?). This would mean that neuroblasts have long temporal windows (again like the mushroom body). If the authors could identify the strategy for the production of neurons, this would be a very significant advance. As it is now, it is difficult to see how the observation can serve as a general principle of high significance.

The authors could, therefore, test whether one NB produces each subtype of if all four contribute to all subtypes. Any neuroblast clone should provide this information (This may have already been figured out by others!) The current method used by the authors will not answer this question since the driver is in INPs but there are drivers for specific individual Type II NBs.

Another approach (not essential): using Notch gof and lof, test whether GMCs produce one or two different neuron(s). By counting NB and INP divisions, this would suggest whether one or all four NB produce the four Cx neurons. This would go a long way to obtain an understanding of the generation of this neural diversity.

2) In Figure 1B, the authors indicate that young neuroblasts express Imp while old neuroblasts express Syp. Then in Figure 2 the authors' image indicates that the CX neurons they are studying are born early during the Imp window. However, Imp and Syp do not act as an ON/OFF switch according to other reports. Without showing that loss of Syp does not lead to the loss of any CX neurons, the authors cannot make this hard cutoﬀ indicated in Figure 2. For example, late-born CX neurons could be born during the Imp/Syp overlap window.

3) Similar to Imp/Syp, the authors break the INP lineage into old and young with D and Ey. However, Grh is a mid-temporal factor for INPs based on the authors' own work. Based on exclusion, the authors indicate that P-EN and P-FN neurons are born from young INPs during the D window. However, the authors need to be careful and either show exactly in which INP temporal window those neurons are born or clearly state that they could either be from D or Grh.

4) The authors need to test whether Ey is sufficient to specify E-PG and PF-R neurons, and if it is sufficient, in which NB temporal window, e.g. during the early or late Imp window.

5) It is not clear from Figure 4 that Toy marks late born CX neurons. Considering that the authors draw a conclusion from Toy staining, they need to clearly show that PF-R and E-PG are Toy+ while P-EN and P-FN are not.

6) The authors need to show that Runt labels only early born neurons in order to use it as a marker. The "data not shown" must be shown in supplemental data.

7) Since the authors show that there are ectopic P-EN and P-FN neurons upon Ey-RNAi, they need to check whether P-FN neurons are also "true" P-FN neurons and that the increased number of neurons detected with the P-FN LexA is not simply labeling additional, non-P-FN neurons.

8) The authors show that PF-R and E-PG neurons are born earlier than P-EN and P-FN neurons along the neuroblast temporal axis. Since temporal patterning is also occurring along that axis (through Imp/Syp) it is not clear when additional P-EN and P-FN neurons are produced in the Ey-RNAi experiment. Does the experiment result in additional P-EN/P-FN neurons during the early or late Imp temporal window? In other words, does the neuroblast temporal axis influence the INP temporal axis? Finer temporal controlled experiments with tub-Gal80ts might help address this question. From a developmental perspective, addressing this question could be extremely interesting for the field.

9) Lineage tracing identified and confirmed that four adult CX columnar neurons are generated by type II NBs using a Gal4 driver in specific populations of INPs along with an intersectional approach. Using this tool, the authors also birthdate the four CX columnar subtypes. Although differences in birthdates are apparent, they see a relatively broad distribution of overlapping times for each subtype. The authors suggest that this is due to DM1-DM4 lineages generating neuron subtypes asynchronously. Supportive evidence for this idea would be to image R9D11-Gal4 (INP-Gal4) to check if some INPs are D+ while others are Eye+ at the same time point.

10) Further, they conclude that all subtypes arise 'early' in type II NBs i.e. born in the Imp+ temporal window. Question: is this just based on timing? Are all four columnar neuron subtypes Chinmo+?

11) They show that E-PG and PF-R neurons are born from older INPs that are Ey+ but P-EN and P-FN are born from younger INPs that are D+. This was done using an old-INP specific driver OK107.

Question: does Ey in these INPs meet the conditions for a TTF that they have outlined in the Introduction? This information should be properly referenced.

12) They address the role of Ey in determining the cell fate of E-PG and PF-R neurons. They use an eyeless enhancer-Gal4 to knock down Ey (RNAi). They found that there was a near complete loss of E-PG and PF-R neurons.

Question: this phenotype could also occur if the enhancer-Gal4s they are using to identify the E-PG and PF-R neurons are regulated by Ey, is there a way to address this?

13) They then suggest that these neurons are transformed into an early-born INP progeny identity. Transformed into what? Although plausible, without knowing what neuron subtypes early INPs generate from those specific NB temporal windows, it seems premature to suggest there is a transformation.

14) They performed an antibody screen and identified that old INP progeny express Toy (PF-R and E-PG) while young INP progeny express Runt (P-EN and P-FN). They should show Runt expression especially because their Eye RNAi could only affect enhancer expression and Toy and Runt are their only readouts for old vs. young progeny number. When they knockdown Ey in old INPs they see a decrease in Toy expressing neurons and an increase in Runt expressing neurons.

15) There seem to be typos in Figures 4 and 5 as they have wrongly labeled Toy and Runt on the Y-axis of panels I. Panels C, F in Figures 4 and 5 appear to be different channels from the same confocal images. This should be clarified and it would be helpful to show a merge of Toy and Runt expression in supplementary data to show that they do not overlap.

16) They show that there is an increase in P-EN and P-FN neurons when they knockdown Ey in old INPs (Figure 5E). This is suggestive that the younger INP progeny type is increased but still is not definitive that there is a transformation of old INP progeny to young INP progeny. The authors should be clearer when they use the term 'transformation' – are they defining an extension of the young INP temporal window as a transformation of cell-fate? Should 'transformation' be applicable to all shifts of temporal windows?

17) The authors show that ectopic P-EN and P-FN neurons show proper targeting. At the end of this paragraph (second paragraph subsection “Loss of Eyeless produces ectopic P-EN neurons with normal morphology and anatomical connectivity”) they say that temporal identity does not determine 'time of neuronal birth'. This sentence is confusing. It seems that when they knockdown Ey they are extending the young INP window and in that case Ey certainly controls the timing of neuronal birth.

18) In behavioral experiments, the authors use a restricted 24 hr temporal shift to affect the production of only the E-PG neurons. Have the authors confirmed that this genetic manipulation actually causes a loss of E-PG neurons?

---

## [Author Response]

Essential revisions:1) The tracing of most if not all CX columnar neurons to TypeII NB lineages has already been shown by the Lee and Reichert labs.

We agree, and now more clearly acknowledge the pioneering work of (Yang et al., 2013) and (Riebli et al., 2013) in the Introduction. Both papers show that there are only four neuroblasts in the entire brain that generate small field columnar neurons innervating the PB, which includes all four neuronal subtypes we study (PF-R, E-PG, P-EN, P-FN). Note that Yang names the neuroblasts DM1-DM4, whereas Riebli names them DM1-DM3 and DM6. We use the DM1-DM4 nomenclature for simplicity.

To go one step further, the authors must trace specific CX neuron types (PF-R, E-PG, P-EN, P-FN) to a specific Type II neuroblast. Since many neurons (30-50) of each type are produced, which neuroblasts produce them and how do they do it? Does each neuroblast produce an early INP and a late INP neuron type? But then, how is it possible to produce 50 neurons? Do GMCs produce two identical neurons, or are they distinct by Notch activity? Does this mean that each neuroblast produces identical INPs for multiple early time windows? Alternatively, as in the case of the mushroom body, all four neuroblasts might act identically and each could produce the same early INPs a dozen times, and each INP would produce four neurons (two D early neurons, maybe as two hemi-lineages, and two Ey late neurons?). This would mean that neuroblasts have long temporal windows (again like the mushroom body). If the authors could identify the strategy for the production of neurons, this would be a very significant advance. As it is now, it is difficult to see how the observation can serve as a general principle of high significance.The authors could, therefore, test whether one NB produces each subtype of if all four contribute to all subtypes. Any neuroblast clone should provide this information (This may have already been figured out by others!) The current method used by the authors will not answer this question since the driver is in INPs but there are drivers for specific individual Type II NBs.

The reviewer is correct, the DM1-DM4 neuroblast clone data show that each of these four neuroblasts generates intrinsic columnar neurons of the central complex targeting the PB, which include all four neuronal subtypes studied here (PF-R, E-PG, P-EN, P-FN). This is now shown in Figure 1A. Thus, the 50 P-FN neurons develop from 8 neuroblasts (bilateral DM1-DM4) or ~6 neurons per neuroblast. These 6 neurons could arise from the first three GMCs in a single INP lineage or perhaps from a single GMC in three INP lineages. We mention this calculation in the Discussion. “Thus, per brain there are 8 parental neuroblasts that generate 30-50 neurons of each subtype, or 4-6 neurons per neuroblast. These 4-6 neurons could arise from 2-3 GMCs in a single INP lineage, or as 1 neuron from 6 different INPs; twinspot MARCM would be needed to determine their precise cell lineage.”

Another approach (not essential): using Notch gof and lof, test whether GMCs produce one or two different neuron(s). By counting NB and INP divisions, this would suggest whether one or all four NB produce the four Cx neurons. This would go a long way to obtain an understanding of the generation of this neural diversity.

We agree, but Notch^intra^ reverts INPs to neuroblasts to create a brain tumor phenotype (Farnsworth et al., 2015; Figure 1J), precluding interpretation of any sibling neuron phenotype, and there are no GMC-specific driver lines available. We are glad the reviewer views these as “not essential” experiments!

2) In Figure 1B, the authors indicate that young neuroblasts express Imp while old neuroblasts express Syp. Then in Figure 2 the authors' image indicates that the CX neurons they are studying are born early during the Imp window. However, Imp and Syp do not act as an ON/OFF switch according to other reports. Without showing that loss of Syp does not lead to the loss of any CX neurons, the authors cannot make this hard cutoﬀ indicated in Figure 2. For example, late-born CX neurons could be born during the Imp/Syp overlap window.

We agree, and thus we have changed to using Ecdysone Receptor (EcR) expression to reveal a binary subdivision of the neuroblast temporal axis. EcR is off ~0-60h ALH and on 60-120h ALH in all type II neuroblasts (Syed et al., 2017). The text and Figure 1 have been updated.

3) Similar to Imp/Syp, the authors break the INP lineage into old and young with D and Ey. However, Grh is a mid-temporal factor for INPs based on the authors' own work. Based on exclusion, the authors indicate that P-EN and P-FN neurons are born from young INPs during the D window. However, the authors need to be careful and either show exactly in which INP temporal window those neurons are born or clearly state that they could either be from D or Grh.

We agree that we previously did not explain our reasoning for excluding Grh. All DM1-DM4 lineages make the PF-R, E-PG, P-EN, P-FN neurons described here, but Grh is not expressed in the DM1 lineage (Bayraktar and Doe, 2013), and thus Grh cannot be a relevant TTF for specifying these CX neuronal subtypes. Moreover, Grh-LOF does not alter the number of Runt+ and Toy+ neurons. We have added this to the Discussion: “We tested Dichaete and Grainyhead for a role in specification of early INP-derived P-EN and P-FN neurons, but observed no phenotype (data not shown); this is unsurprising for Grainyhead, because it is not expressed in the DM1 lineage (Bayraktar and Doe, 2013) which generates P-EN and P-FN neurons.”

4) The authors need to test whether Ey is sufficient to specify E-PG and PF-R neurons, and if it is sufficient, in which NB temporal window, e.g. during the early or late Imp window.

We agree this is an excellent and obvious experiment. We performed it in two ways. First, we previously used R9D11-gal4 UAS-Ey UAS-GFP in an attempt to misexpress Ey only in young INPs, but surprisingly we found that Ey translation was repressed in young INPs: GFP but not Ey was detected in young INPs (Farnsworth et al., 2015; Supp Figure 4). More recently, we attempted to permanently express Ey using a flp-out method (R9D11-FLP, actin-FRT-stop-FRT-Gal4 UAS-eyeless), and again observed a surprising result: loss of all four neuronal subtypes, both early INP-derived and late INP-derived. Our interpretation is that this “immortalized” expression of Eyeless in INPs and their progeny is toxic, leading to cell death, although we can’t rule out that Ey transforms all INP progeny into a late-born cell type that we lack markers to detect. We have added this to the Discussion (paragraph three of subsection “Specification of CX columnar neurons”).

5) It is not clear from Figure 4 that Toy marks late born CX neurons. Considering that the authors draw a conclusion from Toy staining, they need to clearly show that PF-R and E-PG are Toy+ while P-EN and P-FN are not.

We are happy to comply and have now included a new Supplemental Figure (Figure 5—figure supplement 1) showing that PF-R and E-PG are Toy+ Runt- while P-EN and P-FN are Toy- Runt+.

6) The authors need to show that Runt labels only early born neurons in order to use it as a marker. The "data not shown" must be shown in supplemental data.

We are happy to comply and have now included a new Supplemental Figure (Figure 5—figure supplement 1) showing that PF-R and E-PG are Toy+ Runt- while P-EN and P-FN are Toy- Runt+.

7) Since the authors show that there are ectopic P-EN and P-FN neurons upon Ey-RNAi, they need to check whether P-FN neurons are also "true" P-FN neurons and that the increased number of neurons detected with the P-FN LexA is not simply labeling additional, non-P-FN neurons.

We agree, and have now included Figure 6—figure supplement 1 showing that upon Ey-RNAi, the expanded pool of P-FNs all innervate the FB and NO, identical to endogenous P-FN neurons. The addition of these ectopic P-FN neurons results in an enlarged FB and NO.

8) The authors show that PF-R and E-PG neurons are born earlier than P-EN and P-FN neurons along the neuroblast temporal axis. Since temporal patterning is also occurring along that axis (through Imp/Syp) it is not clear when additional P-EN and P-FN neurons are produced in the Ey-RNAi experiment. Does the experiment result in additional P-EN/P-FN neurons during the early or late Imp temporal window? In other words, does the neuroblast temporal axis influence the INP temporal axis? Finer temporal controlled experiments with tub-Gal80ts might help address this question. From a developmental perspective, addressing this question could be extremely interesting for the field.

This is a good point. These neurons are born prior to 60h ALH, which places them in the high Imp, low Syp window. However, we now use EcR expression to give a binary subdivision of the neuroblast temporal axis. EcR is off ~0-60h ALH and on 60-120h ALH in all type II neuroblasts (Syed et al., 2017). We now clarify that all four columnar neuron subtypes are born in the EcR negative window (new Figure 1).

9) Lineage tracing identified and confirmed that four adult CX columnar neurons are generated by type II NBs using a Gal4 driver in specific populations of INPs along with an intersectional approach. Using this tool, the authors also birthdate the four CX columnar subtypes. Although differences in birthdates are apparent, they see a relatively broad distribution of overlapping times for each subtype. The authors suggest that this is due to DM1-DM4 lineages generating neuron subtypes asynchronously. Supportive evidence for this idea would be to image R9D11-Gal4 (INP-Gal4) to check if some INPs are D+ while others are Eye+ at the same time point.

We appreciate the constructive suggestion. Unfortunately, D+ and Ey+ INPs are co-existing within each INP lineage at all larval stages (see Figure 1C-E; Supplemental Tables 1,2; Supplemental Figure 1 in Bayraktar and Doe, 2013). Thus, this stain would not reveal synchronous or asynchronous DM1-DM4 lineages.

10) Further, they conclude that all subtypes arise 'early' in type II NBs i.e. born in the Imp+ temporal window. Question: is this just based on timing? Are all four columnar neuron subtypes Chinmo+?

Yes, the reviewer is correct it is based on timing compared to published data. We show that all four neuronal subtypes are born prior to 60h ALH. This matches the window when Imp is high, Chinmo is high, Syp is low, and EcR is off (after 60h ALH the expression of each marker is reversed). Thus, our assignment is made using published data from multiple labs (Narbonne-Reveau et al., 2016; Ren et al., 2017; Syed et al., 2017; Zhu et al., 2006). We did not stain adult brains for Chinmo because it has not been established that larval Chinmo+ neuroblasts produce adult Chinmo+ neurons (in our hands, adult Chinmo expression seems much broader than just early neuroblast progeny).

11) They show that E-PG and PF-R neurons are born from older INPs that are Ey+ but P-EN and P-FN are born from younger INPs that are D+. This was done using an old-INP specific driver OK107.Question: does Ey in these INPs meet the conditions for a TTF that they have outlined in the ntroduction? This information should be properly referenced.

We agree, and now cite the work of Bayraktar et al., 2013, showing that Ey is a TTF within INP lineages.

12) They address the role of Ey in determining the cell fate of E-PG and PF-R neurons. They use an eyeless enhancer-Gal4 to knock down Ey (RNAi). They found that there was a near complete loss of E-PG and PF-R neurons.Question: this phenotype could also occur if the enhancer-Gal4s they are using to identify the E-PG and PF-R neurons are regulated by Ey, is there a way to address this?

We use three different Gal4 lines to drive Ey RNAi and only two are Ey enhancer lines, the third is from the D locus. More importantly, we identify the four neuronal subtypes using LexA lines. Thus, even if the Ey RNAi led to loss of Ey enhancer Gal4 expression (e.g. via loss of positive feedback) it should only happen in two of the three Gal4 lines, and it would not lead to silencing of our LexA reporter lines. We have changed the title of this panel to be “LexA reporters for CX columnar neurons” to make this point more clearly.

13) They then suggest that these neurons are transformed into an early-born INP progeny identity. Transformed into what? Although plausible, without knowing what neuron subtypes early INPs generate from those specific NB temporal windows, it seems premature to suggest there is a transformation.

We have added Figure 6—figure supplement 2 to clarify this point: we know where the four subtypes of columnar neurons arise in the type II neuroblast lineage, but the neurons born before or after them remain completely unknown. For the past few years we have looked for additional markers for these neurons, but without success. We would love more markers but we feel our study using two early INP neuron types and two late INP neuron types reveals an important role for temporal identity in specifying CX neuronal subtypes required for navigation.

14) They performed an antibody screen and identified that old INP progeny express Toy (PF-R and E-PG) while young INP progeny express Runt (P-EN and P-FN). They should show Runt expression especially because their Eye RNAi could only affect enhancer expression and Toy and Runt are their only readouts for old vs. young progeny number. When they knockdown Ey in old INPs they see a decrease in Toy expressing neurons and an increase in Runt expressing neurons.

We agree and show Runt expression in Figure 5 and the new Figure 5—figure supplement 1.

15) There seem to be typos in Figures 4 and 5 as they have wrongly labeled Toy and Runt on the Y-axis of panels I. Panels C, F in Figures 4 and 5 appear to be different channels from the same confocal images. This should be clarified and it would be helpful to show a merge of Toy and Runt expression in supplementary data to show that they do not overlap.

You are right – we have made the correction.

16) They show that there is an increase in P-EN and P-FN neurons when they knockdown Ey in old INPs (Figure 5E). This is suggestive that the younger INP progeny type is increased but still is not definitive that there is a transformation of old INP progeny to young INP progeny. The authors should be clearer when they use the term 'transformation' – are they defining an extension of the young INP temporal window as a transformation of cell-fate? Should 'transformation' be applicable to all shifts of temporal windows?

We agree and have dropped all use of the word “transformation” in the text.

17) The authors show that ectopic P-EN and P-FN neurons show proper targeting. At the end of this paragraph (second paragraph subsection “Loss of Eyeless produces ectopic P-EN neurons with normal morphology and anatomical connectivity”) they say that temporal identity does not determine 'time of neuronal birth'. This sentence is confusing. It seems that when they knockdown Ey they are extending the young INP window and in that case Ey certainly controls the timing of neuronal birth.

We agree. We now say: “This shows that neuronal birth-date can be uncoupled from neuronal morphology, because we see P-EN neurons born later than normal in the INP lineage establish morphology that mimics that of the endogenous, early-born P-EN neurons.”

18) In behavioral experiments, the authors use a restricted 24 hr temporal shift to affect the production of only the E-PG neurons. Have the authors confirmed that this genetic manipulation actually causes a loss of E-PG neurons?

Thank you for this comment, which we have used to clarify our manuscript. In fact, we do not expect the E-PG neurons to be gone, as that would lead to massive morphological defects in the CX (see Xie et al., 2017) that we did not observe in our experiment. Indeed, we see no loss of E-PG neurons when we assay them using the E-PG reporter R60D05. We have clarified our text in response to this excellent comment by adding new text to the Results and Discussion.

Results: “Both control and Eyeless^RNAi^ animals exposed to this regime had no major morphological defects in the central complex (Figure 7B,C; data not shown), indicating that E-PG neuron number is likely normal.”

Discussion: “Eyeless^RNAi^ causes no gross deformities in the CX, suggesting E-PGs were not eliminated by Eyeless^RNAi^ using this regime, as loss of all E-PG neurons produces severe EB defects (Xie et al., 2017). The developmental defects in E-PG neurons could be misexpression of ion channels or other functionally important molecules, rather than apoptosis.”